# Structured information extraction from scientific text with large language models

John Dagdelen [1,2,3], Alexander Dunn [1,2,3], Sanghoon Lee[1,2], Nicholas Walker[1], Andrew S. Rosen [1,2], Gerbrand Ceder[1,2], Kristin A. Persson [1,2] & Anubhav Jain [1] ✉

Extracting structured knowledge from scientific text remains a challenging task for machine learning models. Here, we present a simple approach to joint named entity recognition and relation extraction and demonstrate how pre-trained large language models (GPT-3, Llama-2) can be fine-tuned to extract useful records of complex scientific knowledge. We test three representative tasks in materials chemistry: linking dopants and host materials, cataloging metal-organic frameworks, and general composition/phase/morphology/application information extraction. Records are extracted from single sentences or entire paragraphs, and the output can be returned as simple English sentences or a more structured format such as a list of JSON objects. This approach represents a simple, accessible, and highly flexible route to obtaining large databases of structured specialized scientific knowledge extracted from research papers.

The majority of scientific knowledge about solid-state materials is scattered across the text, tables, and figures of millions of academic research papers. Thus, it is difficult for researchers to properly understand the full body of past work and effectively leverage existing knowledge when designing experiments. Moreover, machine learning models for direct property prediction are being increasingly employed as screening steps for materials discovery and design workflows[1–3], but this approach is limited by the amount of training data available in tabulated databases. While databases of materials property data derived from ab initio simulations are relatively common, they are limited to the subset of computationally accessible properties whereas databases of experimental property measurements and other useful experimental data are comparatively small (if they exist at all).

In recent years, researchers have made significant advances in the application of natural language processing (NLP) algorithms for materials towards structuring the existing body of textual materials science knowledge[4–7]. The majority of this work has focused on named entity recognition (NER), where entity labels such as "material" or "property" are applied to words from the text. These tagged sequences of words can sometimes be used with additional post-processing to construct auto-generated tabular databases of materials property data

aggregated from text entries[8–12]. Prior information extraction studies in the domain of solid-state materials include NER labeling of chemical synthesis parameters in methods section texts[13–16], quantitative results of battery cycling experiments[17], or peak absorption wavelengths for UV-Vis experiments[18], among others[4,5,9–12,19]. Regular expressions, BiLSTM recurrent neural networks, and smaller transformer-based language models such as BERT are sufficient for such tasks. In these studies, entities (e.g., LiCoO2, "350K") rather than relations (e.g., "350K" is an experimental synthesis parameter for LiCoO$_2$) are the primary target of extraction.

Yet, a key challenge in scientific natural language processing is the development of robust, simple, and general relation extraction (RE) techniques to accurately extract the relationships between named entities. Downstream tasks such as supervised machine learning or the construction of knowledge graphs require the transformation of unstructured text into sets of structured relationships between semantic entities of interest. RE models are used to determine which entities are linked by a predefined set of relations. For example, in the sentence "LiCoO2 is studied as a Li-ion battery material", the material entity "LiCoO2" is linked to the application entity "Li-ion battery". Until recently, there has been relatively little work on relation extraction in

[1]Lawrence Berkeley National Laboratory, Berkeley, CA, USA. [2]Materials Science and Engineering Department, University of California, Berkeley, CA, USA. [3]These authors contributed equally: John Dagdelen, Alexander Dunn. ✉e-mail: ajain@lbl.gov

materials science text, but there has been much research interest in RE on general-purpose text, especially related to linking people, organizations, locations, and dates[20,21]. These methods have traditionally relied on pipeline-based approaches where named entity recognition is the first step followed by one or more additional steps and a final relation classification step (see Fig. 1, top row). Each of these steps typically uses a separate machine learning model, which may or may not share weights or architectures with each other. State-of-the-art transformer-based implementations of pipeline implementations have been shown to perform document level relation extraction relatively well on a variety of general-knowledge corpora[22] and more specialized domains such as chemical-disease relations[23] and gene-disease relations[24]. Recently, this kind of two-step approach was demonstrated on a benchmark dataset of procedures for the synthesis of polycrystalline materials encoded as directed graphs extracted from materials science text[25].

However, scientific information often cannot be modeled as simple pairwise relations between entities. This is particularly apparent in inorganic materials science, where a compound's properties are determined by a complex combination of its elemental composition, atomic geometry, microstructure, morphology (e.g., nanoparticles, heterostructures, and interfaces), processing history, and environmental factors such as temperature and pressure. Furthermore, inorganic materials knowledge is often inherently intertwined such that the relations may only be valid between one entity type and a compound entity (itself comprised of several entities and relationships). For example, we may consider zinc oxide nanoparticles (a composition "ZnO" linked to the morphology "nanoparticles") to be a catalyst, but "ZnO" and "nanoparticles" alone are not necessarily catalysts in

themselves. When parts of these compound relations are lost, scientific meaning will change. A sample of an "epitaxial La-doped thin film" of $HfZrO_4$ will have different physical properties than a "La-doped thin film" of $HfZrO_4$ and a "La-doped" sample of $HfZrO_4$. In theory, relationships between $n$ entities can be modeled as $n$-tuples (e.g., ("ZnO", "nanoparticles", "catalyst")), but comprehensively enumerating all possible variations is both impractical and not amenable to conventional relation extraction methods, since a sufficient number of training examples is required for each relation type. For example, a model extracting 10 distinct entity classes may have $^{10}C_3 = 120$ 3-tuple entity relation types, each requiring at least several annotation examples. Current relation extraction models are not designed to practically extract or preserve such kinds of highly complex, intricately related, and hierarchical relationships between arbitrary numbers of named entities; a more flexible strategy is required.

Large language models (LLMs) such as GPT-3/4[26,27], PaLM[28], Megatron[29], LLaMA 1/2[30,31], OPT[32], Gopher[33], and FLAN[34] have been shown to have remarkable ability to leverage semantic information between tokens in natural language sequences of varying length. They are particularly adept at sequence-to-sequence (*seq2seq*) tasks, where text input is used to seed a text response from the model. In this paper, we will refer to these inputs as "prompts" and the outputs as "completions." Use cases for seq2seq are broad[35] and include machine translation[36], answering general factual knowledge questions[33,37], performing simple arithmetic[33], translating between languages[36,38], summarizing text[28,39], and chatbot applications[26,40]. It stands to reason that these models may also be adept at complex scientific information extraction.

Recently, end-to-end methods that use a single machine learning model have been investigated for joint named entity recognition and

**(a) Previous**: Multi-step (pipeline) named entity recognition and relationship extraction

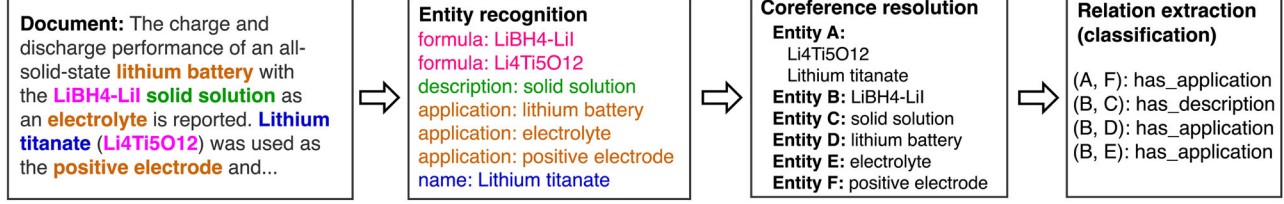

**(b) Previous**: seq2seq enumerates relationships as 2-tuples

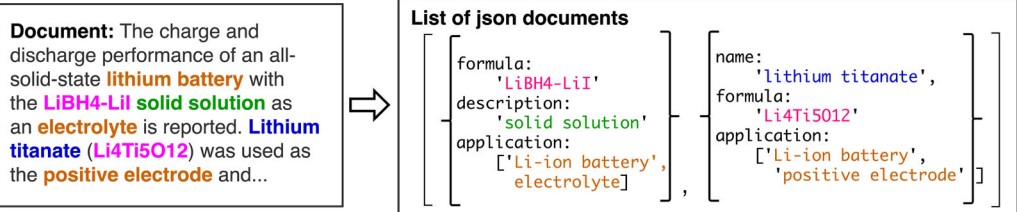

**(c) This work:** Hierarchical entity relationships without explicit enumeration

**Fig. 1 | Schematic comparison of previous relation extraction (RE) methods to this work.** The objective of each method is to extract entities (colored text) and their relationships from unstructured text. **a** An example multi-step pipeline approach first performs entity recognition, then intermediate processing such as coreference resolution, and finally classification of links between entities. **b** seq2seq approaches encode relationships as 2-tuples in the output sequence. Named entities and relationship links are tagged with special symbols (e.g., "@FORMULA@", "@N2F@"). **c** The method shown in this work outputs entities and their relationships as JSON documents or other hierarchical structures.

relation extraction (NERRE)[41–43]. These methods take a sequence-to-sequence approach where a model is trained to output tuples of two or more named entities and the relation label belonging to the predefined set of possible relations between them (Fig. 1, middle row). These methods perform well on relation extraction, but they fundamentally remain $n$-ary relation extraction systems that are not suited to highly intricate and hierarchical NERRE.

In the domain of materials science, Huang & Cole recently fine-tuned a BERT model on battery publications and trained a model to enhance a database of NLP-extracted battery data[11]. Their approach employed a "question and answer" (Q/A) approach that extracted limited device-level information (e.g., "What is the cathode?", "What is the anode?", "What is the electrolyte?") in tandem with conventional information extraction methods[11]. We note that this approach cannot be used on passages that contain information about more than one device, and it required the BERT language model to be trained on hundreds of thousands of battery research papers before being fine-tuned on the Q/A task. More recently, Zheng et al.[44] designed a prompt-engineering approach (ChemPrompt w/ ChatGPT[45]) for extracting data from scientific papers. This method is focused on structuring text into tabular forms, creating semi-structured summaries, and collating existing knowledge from the pretraining corpus. Similarly, Castro Nascimento and Pimentel[46] examined ChatGPT's general knowledge of chemistry; however, they find that, as opposed to methods using considerable prompt engineering[47], ChatGPT without prompting "tricks" performs poorly on several simple tasks in chemistry. Xie et al.'s[48] approach utilizes LLMs fine-tuned on a large, broad materials science corpus for a range of Q/A, inverse design, classification, and regression tasks. While these methods[44,46–48] demonstrate LLMs might act as materials science knowledge engines, they have not been shown to extract structured representations of complex hierarchical entity relationships generalizing outside of the pretraining corpus.

In this work, we investigate a simple approach to complex information extraction where a large language model is fine-tuned to simultaneously extract named entities and their relationships. This method is able to flexibly handle complex inter-relations (including cases where information exists as lists of multiple items) without requiring enumeration of all of possible $n$-tuple relations or preliminary NER. Our approach differs from the supervised learning (e.g., regression and classification for chemistry) and inverse design approaches of Jablonka et al.[49,50] and Xie et al.[48]; rather than using LLMs to directly influence design or predict properties, we aim to (accurately) extract structured hierarchies of information for use with downstream models. We fine-tune a pretrained large language model (e.g., GPT-3[26] or Llama-2[31]) to accept a text passage (for example, a research paper abstract) and write a precisely formatted "summary" of knowledge contained in the prompt. This completion can be formatted as either English sentences or a more structured schema such as a list of JSON documents. To use this method, one only has to define the desired output structure—for example, a list of JSON objects with a predefined set of keys—and annotate ~100–500 text passages using

this format. The LLM is then fine-tuned on these examples, and the resulting model is able to accurately output extracted information in the same structured representation, such as the format shown in Fig. 1. In essence, a domain expert can show an LLM-NERRE model both what it should extract and how that information should be represented, and then the model learns how to perform the task independently.

This method shows strong performance using both OpenAI's GPT-3 (closed source) and Llama-2 (open access) on both sentence-level and document-level materials information extraction. Moreover, the method can leverage online LLM APIs, which allows users to train bespoke models without extensive knowledge of how LLMs work internally; the LLM may be simply treated by the user as a black-box that transforms passages into precisely-formatted, structured summaries of scientific text. Therefore, researchers may use this method with little NLP experience. We also discuss how intermediate models can be used to pre-suggest entities for annotation, vastly increasing the speed and ease of annotating documents so that large training sets can be constructed relatively quickly. Although the example tasks shown are from materials science, the generality and accessibility of the method implies it may be readily applied to other domains such as chemistry, health sciences, or biology. In particular, this approach does not appear to require fine-tuning on a large corpus of domain-specific data (e.g., millions of article abstracts or paragraphs) as in previous methods; rather, the comprehensive pretraining of the LLMs along with the user-provided annotations are sufficient to accomplish a broad array of complex tasks.

## Results

We use the described approach on three joint named entity recognition and relation extraction (NERRE) materials information extraction tasks: solid-state impurity doping, metal–organic frameworks (MOFs), and general materials information extraction. Details for each dataset are summarized in Table 1. Further details of each task are presented in the Methods section. Briefly, the solid-state impurity task is to identify host materials, dopants, and potentially additional related information from text passages (sentences). The MOF task is to identify chemical formulae, applications, guest species, and further descriptions of MOF materials from text (materials science abstracts). The general materials information task is to identify inorganic materials, their formulae, acronyms, applications, phase labels, and other descriptive information from text (materials science abstracts). The general and MOF models were trained on data including normalization and error correction, while doping models were trained to extract data exactly as it appears in text. Each base LLM model is fine-tuned per-task to adhere to a particular schema that encapsulates the entities of interest, relevant relationships, and format. All schemas are shown in Table 1 and further details are available in the Methods and Supplementary Note 1.

### Relation extraction performance

A comparison between GPT-3 and Llama-2 on NERRE precision, recall, and $F_1$ scores across the three tasks using a JSON schema is shown in Table 2. Details on each of the task's JSON schemas are explained in the Methods section. The performances are calculated with an exact word-

**Table 1 | Overview of approaches tested on the three materials information extraction tasks**

| Task | Schema | Training samples | Task level | Completion format |
|------|--------|------------------|------------|-------------------|
| Doping | `Doping-JSON` | 413 sentences | Sentence | JSON |
| Doping | `Doping-English` | 413 sentences | Sentence | English sentences |
| Doping | `DopingExtra-English` | 413 sentences | Sentence | English sentences |
| MOFs | `MOF-JSON` | 507 abstracts | Abstract | JSON |
| General Materials | `General-JSON` | 634 abstracts | Abstract | JSON |

All three tasks are tested with a JSON schema, and we additionally test the doping task with alternate schemas resembling written English. The MOF and general materials models are trained and evaluated on abstracts, while doping tasks are evaluated on sentences.

**Table 2 | Named entity recognition and relation extraction scores for three tasks in materials science using models with a JSON output schema**

| Task | Relation | E.M. Precision (GPT-3) | E.M. Recall (GPT-3) | E.M. $F_1$ (GPT-3) | E.M. Precision (Llama-2) | E.M. Recall (Llama-2) | E.M. $F_1$ (Llama-2) |
|------|----------|------------------------|---------------------|--------------------|--------------------------|-----------------------|----------------------|
| Doping | host-dopant | 0.772 | 0.684 | 0.726 | 0.836 | 0.807 | **0.821**[a] |
| General | formula-name | 0.507 | 0.429 | **0.456** | 0.462 | 0.417 | 0.367 |
| General | formula-acronym | 0.500 | 0.250 | **0.333** | 0.333 | 0.250 | 0.286 |
| General | formula-structure/phase | 0.538 | 0.439 | **0.482** | 0.551 | 0.432 | 0.47 |
| General | formula-application | 0.542 | 0.543 | **0.537** | 0.545 | 0.496 | 0.516 |
| General | formula-description | 0.362 | 0.35 | **0.354** | 0.347 | 0.342 | 0.340 |
| MOFs | name-formula | 0.425 | 0.688 | **0.483** | 0.460 | 0.454 | 0.276 |
| MOFs | name-guest specie | 0.789 | 0.576 | **0.616** | 0.497 | 0.407 | 0.408 |
| MOFs | name-application | 0.657 | 0.518 | **0.573** | 0.507 | 0.562 | 0.531 |
| MOFs | name-description | 0.493 | 0.475 | **0.404** | 0.432 | 0.411 | 0.389 |

Exact match (E.M.) scores are evaluated on a per-word basis, and links are only correct if both entities and the relationship are correct. The exact match metric scores output that contains the correct information but is written differently as incorrect, making such scores a rough lower bound on the true performance of models. $F_1$, precision, and recall reflect the scores on a hold out test set for doping models and averages over five cross-validation sets for the general and MOF models.
[a]Best $F_1$ scores for each task are shown in bold.

match basis, a lower-bound metric described in more detail in the Methods section. Because this joint task involves both named-entity recognition and relation extraction, it reflects both the NER and RE performance of the models (a relation cannot be correctly identified if the entities are not correct.) GPT-3 achieves the highest $F_1$ scores for the General and MOF tasks across all the entity relationships we tested. Exact match $F_1$ scores for these two extraction tasks are generally ~30% lower than in the host-dopant task. The highest $F_1$ for the general task is found for relationships between formulae and applications ($F_1 = 0.537$) while formula-acronym and formula-description relationships are much less reliable. A similar finding occurs for the MOF task, where the name-application ($F_1 = 0.573$) and name-guest species ($F_1 = 0.616$) relationships are extracted most accurately. The Llama-2 NERRE scores are on average $20-30\%$ lower than their GPT-3 counterparts, indicating a significant advantage for GPT-3. In the dopant task, Llama-2 has the highest precision (0.836), recall (0.807), and $F_1$ (0.821), representing an improvement of 13% over GPT-3 wrt. $F_1$.

The $F_1$ scores for the general and MOF tasks in Table 2 are generally 0.3–0.6, which is, on first inspection, seemingly too low to be useful for a large scale information extraction task. However, the scores for the MOF and general tasks are subject to an important caveat. These tasks' annotations include implicit normalization (e.g. "Lithium ion" → "Li-ion") and error correction ("silcion" → "silicon"), while the doping task aims to extract hosts and dopants exactly as they appear in text. Thus, the exact word-match basis scores shown above are an approximate lower bound on information extraction performance, since this metric compares only exact matches between words. When outputs of the general and MOF models are read by human experts, it becomes obvious that the models are often extracting true information with slight changes in phrasing or notation. There is also an effect on performance from inherent ambiguity in real-world information extraction tasks. For example, in MOF information extraction, MOF names (e.g., "ZIF-8") are qualitatively easier to delimit than descriptions (e.g., "mesostructured MOFs formed by Cu2+ and 5-hydroxy-1,3-benzenedicarboxylic acid"), which can be written with many different wordings.

To account for these factors, we manually scored outputs against the original human (true) annotations for a random 10% test set of the general materials information extraction dataset. We calculated "manual scores" by marking extractions as correct if the core information from entities is extracted in the correct JSON object (i.e., grouped with the correct material formula) and incorrect if they are in the wrong JSON object, are not extracted at all, or are not plausibly inferred from the original abstract. In contrast to the exact match

**Table 3 | Manual scores for the general materials task using GPT-3 with General-JSON schema**

| Entity | Extraction recall | Extraction precision | Extraction $F_1$ |
|--------|-------------------|----------------------|------------------|
| formula | 0.943 | 0.943 | 0.943 |
| name | 0.692 | 1.0 | 0.818 |
| acronym | 0.667 | 0.400 | 0.500 |
| applications | 0.797 | 0.870 | 0.832 |
| structure or phase | 0.754 | 0.920 | 0.829 |
| description | 0.576 | 0.905 | 0.704 |

Scores measure the model's ability to extract inter-related data together (i.e. assigning entities correct labels and grouping them appropriately).

scores (Table 2), manual scores allow for flexibility with respect to three aspects: (1) entity normalization, (2) error correction, and (3) multiple plausible annotations of an entity under different labels (e.g., "thermoplastic elastomer" may be considered either an application or description). Whereas Table 2 assesses whether the model can extract pairs of words exactly as they appear in the true annotation, the manual scores shown in Table 3 assess if the model extracts equivalent information to that of the true annotation - regardless of the exact form. Simply, if a domain expert would agree the model's extraction and the true extraction are equivalent, the model's extraction is marked as correct. We provide precise details on this procedure in the Methods section and detailed examples with explanations in Supplementary Discussion 4.

Table 3 shows the adjusted scores based on manual scoring. We stratify these scores by entity; the "name", "acronym", "application", "structure", and "description" manual scores can be compared to Table 2's exact-match `formula-{name, application, structure, description}` relation scores. For example, "description" reflects how often the model extracts a description entity which is both equivalent in meaning to that of the true annotation (according to a domain expert) and is grouped in the correct JSON object (linked to the correct formula). We see that exact-match scoring severely under-predicts performance for materials' names (0.456 vs 0.818), applications (0.537 vs 0.832), structures/phases (0.482 vs 0.829), and descriptions (0.354 vs 0.704). Manual scoring reveals that our models are actually able to correctly extract structured knowledge from scientific text on a wide variety of materials science topics, and readers can inspect the model's output on test set examples (included in the Supplementary Discussion 4) for themselves. We observe that

acronyms have the lowest information extraction scores, which we attribute to the fact that acronyms are relatively rare in the training class compared to the others (appearing in only 52 abstracts across the entire dataset, ~9% of the documents) and that the model can confuse acronyms with chemical formulae (e.g., "AuNP" is the acronym for gold nanoparticle but is also a valid chemical formula). Usually, context clues are the only way to disambiguate cases like this, and we expect including more training data with acronyms may improve the acronym extraction score.

Overall, these scores indicate the model is highly capable at extracting meaningfully complex representations of material knowledge from passages. Precision scores for the various categories (other than acronyms) are all roughly 0.87 or better, which indicates that when information is extracted, it contains true relational information from the passage rather than spurious connections.

The advantage of the LLM-NERRE method reflected in these manual scores is the ability to automatically correct errors and normalize common entity patterns. While the doping models were trained to extract snippets of text exactly as they appeared in the text prompt, the General-JSON model's training data included simple normalizations and error corrections of entities. For example, the erroneous inclusion of white spaces in chemical formulae is common in the raw abstract text. We observe that including corrected formulae instead of the raw string in the output training sequences results in LLMs that automatically resolve extracted entities to cleaner forms. For example, "Li Co O2" is corrected to "LiCoO2" by the model without additional post-processing. Similarly, because there are sufficient training examples, the models using `General-JSON` schema resolve text such as "PdO functionalized with platinum" to a normalized form such as `{formula: "PdO", description: ["Pt-functionalized"]}`. The built-in normalization and correction abilities of LLM models may prove useful for domain specialists who desire structured entity formats rather than exact string excerpts pulled directly from the text, as entity normalization is a common post-processing task.

### Effect of different schemas

For the host-dopant extraction task, we evaluated three different output schemas to determine whether one format of output is exclusively better than any other. The models using the Doping-English schema output English sentences with a particular structure (e.g., "the host '<host entity>' was doped with '<dopant entity>'.") and the DopingExtra-English models likewise output English sentences but also includes some additional information (e.g., if one of the hosts is a solid solution and/or the concentration of a particular dopant). For the Doping-JSON schema, we used a JSON object schema with keys "hosts", "dopants", and "hosts2dopants" (which in turn has a key-value object as its corresponding value). For readers familiar with the Python programming language, these are identical to python dictionary objects with strings as keys and strings or other dictionaries as values. We include a baseline comparison to seq2rel[41], a comparable sequence-to-sequence method, trained on the same doping dataset. We also compare to MatBERT-Doping[5], an NER model trained on ~450 abstracts, combined with a simple heuristic for determining host-dopant relationships; that is, all hosts and dopants within the same sentence (sample) are related. We refer to this model as MatBERT-Proximity. Full descriptions and examples of all schemas are available in the Methods section, and further details on seq2rel and MatBERT-Proximity are available in Supplementary Notes 4–5. Because the general materials information extraction and MOF information extraction tasks are far more complex, we did not attempt to train models to output English sentences (as opposed to JSON formatted strings), as the resulting sequences would be difficult to parse into structured database entries.

We find that all three of our LLM-NERRE host-dopant extraction models perform significantly better than either the MatBERT-Proximity or seq2rel baseline models. Of the two baselines, the

seq2rel model achieves higher precision (0.420) and recall (0.605) resulting in $F_1 = 0.496$, which is slightly higher than MatBERT-Proximity (0.390) but substantially lower than any of the LLM-NERRE models. This seq2rel benchmark model is derived from the PubMedBERT[51] pretrained BERT model as per the original implementation[41], and it may be possible to improve the seq2rel method by using a BERT model pretrained exclusively on materials text rather than biomedical text. However, this improvement is not expected to be dramatic because previous comparisons between SciBERT and MatBERT show relatively minor differences in materials NER tasks[5]. We also observe that all three LLM-NERRE models exceed the performance of the two baselines in pure NER performance (see Supplementary Discussion 2) despite being trained on less text than the MatBERT-NER model (413 sentences vs. 455 abstracts.) Of the six LLM-based models, the Llama-2 model with Doping-JSON schema performs the best ($F_1 = 0.821$) with GPT-3/DopingExtra-English ($F_1 = 0.809$) and Llama-2/Doping-English ($F_1 = 0.814$) both within a 2% margin. We summarize both LLMs' performances with all three schemas alongside the baseline models in Table 4.

Within the GPT-3 results, the DopingExtra-English and Doping-English schemas have the highest $F_1$. In particular, GPT-3/DopingExtra-English tops the GPT-3 models despite being trained on the same number of samples as the Doping-English and Doping-JSON models. This is notable because GPT-3/DopingExtra-English is both more accurate and more capable (i.e., this model extracts "results" and "modifiers" entities in addition to host-dopant relationships) than the GPT-3 models using other schemas. The opposite observation is true of the Llama-2 models, where the JSON format outperforms both English schemas and the DopingExtra-English schema suffers from low precision (0.694). Roughly, the GPT-3 models tend to perform optimally when using natural language like schemas, while Llama-2 performs optimally using JSON.

### Human-in-the-loop annotation

As a separate experiment, we evaluated the use of partially trained LLMs in a "human-in-the-loop" annotation process for constructing outputs with the GPT-3/General-JSON, as seen in Fig. 2. In each trial of the experiment, the human annotator received 10 abstracts and 10 schemas that were pre-populated by an intermediate version of the model which was trained on $n$ samples of training data ($n = 1, 10, 50, 100, 300$). Instead of completing annotations from scratch, the human annotator corrected these intermediate models' suggestions, and the time to complete each annotation was recorded. As shown in Fig. 3, the annotation time sharply decreases as the number of training samples used in the intermediate models increases; the $n = 300$ intermediate model was able to reduce the average annotation time per abstract by 57% in comparison with the $n = 1$ model, indicating that the model was completing many sections of the annotation correctly.

At low numbers of training samples, the models' predictions are not valid JSON objects, and the annotator had to redo annotations from scratch. At higher numbers of training samples, particularly those above 50, the intermediate model predictions required very little error correction from the annotator. As a lower bound, we also report the time needed by the annotator to simply verify whether an entry was entirely correct (verification time) which reflects the annotation rate of a human annotator using a perfect model, which only requires the human annotator to check the outputs. We find that by three metrics (time per abstract, time per material entry, and time per prompt token), the human annotator annotated substantially faster with a well-trained model in the loop ($n$ samples > 50) than with a poorly trained model ($n$ samples < 50) or no model. For example, the $n = 300$ model reduced the annotation time per token ~60% compared to the $n = 1$ model and is only 38% slower than the verification time. Given additional training samples for intermediate models, we expect the annotation to asymptotically approach the verification time. Thus, this

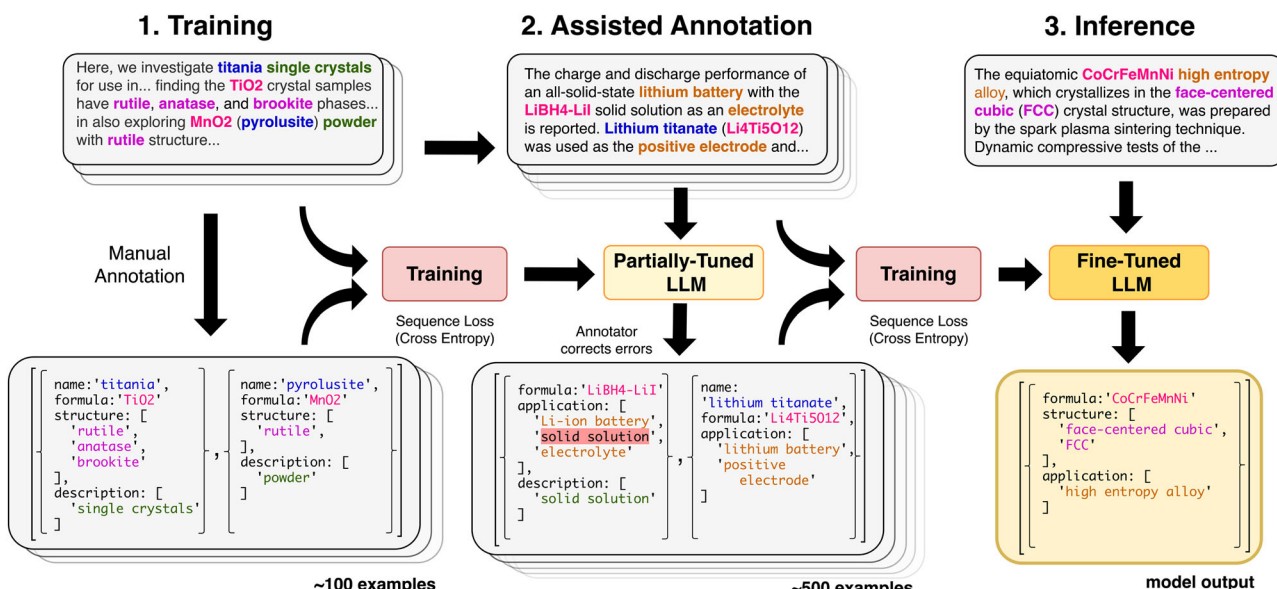

**Fig. 2 | Overview of the proposed sequence-to-sequence approach to document-level joint named entity recognition and relationship extraction task.** In the first step, lists of JSON documents are prepared from abstracts according to a predefined schema, and the large language model (LLM) is trained. In the second step, this preliminary (intermediate) model is used to accelerate the preparation of additional training data by pre-annotation with the partially trained model and manual correction. An example error is shown high-lighted in red. This step may be repeated multiple times with each subsequent partial fine-tuning improving in performance. In the final step, the LLM is fine-tuned on the complete dataset and used for inference to extract desired information from new text.

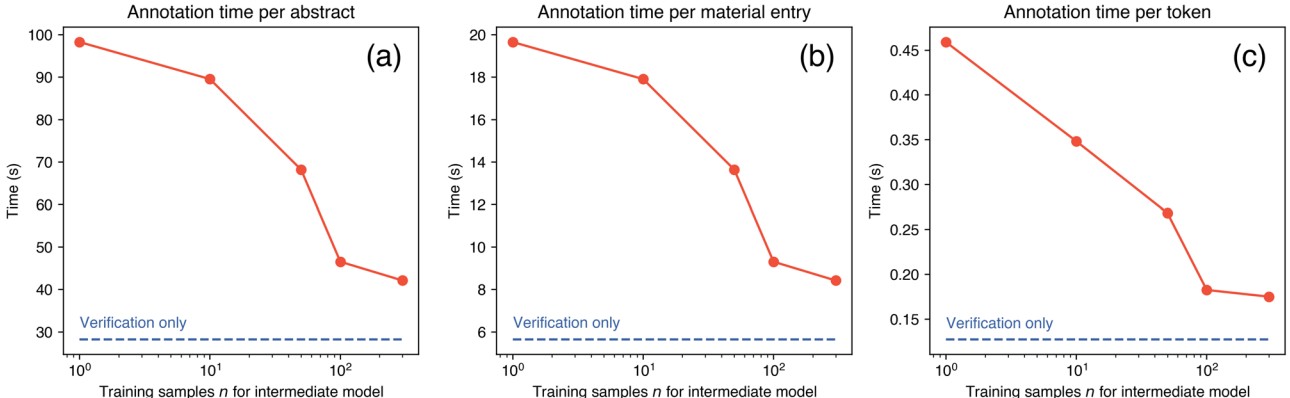

**Fig. 3 | Annotation time as a function of intermediate large language model (LLM) fine-tuning samples for the named entity recognition and relation extraction (NERRE) method.** We show the time taken for a domain expert to annotate new abstracts for the general materials chemistry task with assistance from intermediate (partially-trained) LLM-NERRE models on a (**a**) word basis, (**b**) material entry basis, and (**c**) token basis. Outputs from models trained on more data contain fewer mistakes and require less time to correct. Source data are provided as a Source Data file.

method may serve as a useful tool for building even larger benchmark datasets for information extraction tasks.

We note that the LLM-NERRE method requires the model to learn both the correct structure of the output data as well as the information to populate into that data structure, particularly when asking the model to output English sentences that can later be parsed to a structured format. To determine the minimum number of training examples required for models with sentence-format outputs that have a parseable sentence structure, we trained intermediate models on varying training set sizes for the GPT-3/Doping-English model. Precision, recall, and $F_1$ scores as a function of training set size are plotted in Fig. 4. We observe that output sequences are not properly structured for training set sizes below ~10 samples, but there is a sharp increase in the number of correctly structured outputs at ~20 samples, which seems to be the minimum number of examples GPT-3 needs to learn a desired output format when using simple sentence-type schemas.

## Discussion

Overall, we find excellent performance on three diverse tasks for materials science and engineering: solid-state impurity doping, metal–organic frameworks, and general materials relations. The non-technical nature of this approach implies scientists without NLP training can utilize existing models such as GPT-3 to extract large structured relational datasets for highly-specific problems. As the LLM is treated essentially as a black-box, we anticipate this approach may be used for LLMs other than GPT-3 or Llama-2, including LLMs released in the near future. We hope this approach enables domain specialists to rapidly extract relational datasets for the advancement of scientific knowledge.

The NERRE scores in Tables 2–4 provide a quantitative score for performance, but some of the best features of this method are not directly shown by $F_1$ scores. The primary advantage of this method is its accessibility and ease-of-use, as LLM-NERRE requires only

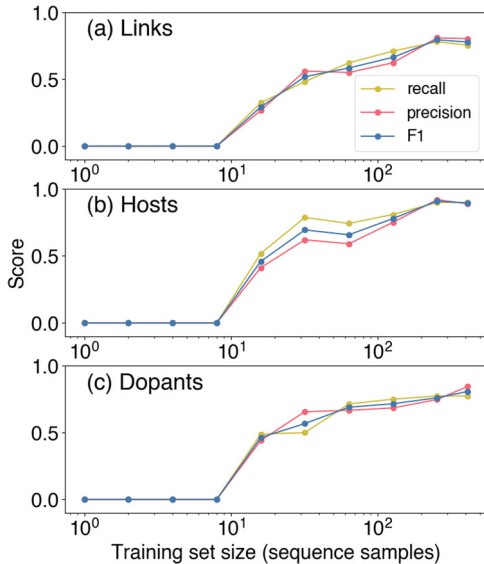

**Fig. 4 | Test set performance vs. number of training samples for the doping extraction task using GPT-3 with the Doping-English schema.** This schema specifically requires the model to learn a new and specific sentence structure to use as the output. We separate scores by (**a**) host-dopant links (relations), (**b**) host entities alone, and (**c**) dopant entities alone. We note that below approximately 10 samples, the scores are zero because the model has not learned the specific structure of the desired output sentences. Source data are provided as a Source Data file.

**Table 4 | Comparison of large language models with different joint named entity recognition and relation extraction (NERRE) schemas to baseline models on host-dopant extraction task**

| Model | Schema | Precision (exact match) | Recall (exact match) | $F_1$ (exact match) |
|---|---|---|---|---|
| MatBERT-Proximity | n/a | 0.377 | 0.403 | 0.390 |
| Seq2rel | n/a | 0.420 | 0.605 | 0.496 |
| GPT-3 | `Doping-JSON` | 0.772 | 0.684 | 0.725 |
| GPT-3 | `Doping-English` | 0.803 | 0.754 | 0.778 |
| GPT-3 | `DopingExtra-English` | 0.820 | 0.798 | 0.809 |
| Llama-2 | `Doping-JSON` | **0.836**[a] | 0.807 | **0.821** |
| Llama-2 | `Doping-English` | 0.787 | **0.842** | 0.814 |
| Llama-2 | `DopingExtra-English` | 0.694 | 0.815 | 0.750 |

NERRE exact match scores are evaluated on a per-word basis, and links are only correct if both entities and relationship are correct. DopingExtra-English scores here refer to only host-dopant relation prediction. We note that exact match scores output that contains the correct information but is written differently as incorrect, making such scores an approximate lower bound on the true performance of models. $F_1$, precision, and recall are computed on a hold-out test set from 77 sentences. Best scores for precision, recall, and $F_1$ are shown in bold.
[a]Best scores among all models in each category (exact match precision, recall, $F_1$) are shown in bold.

specifying a basic schema, annotating a minimal number of examples, and fine-tuning a model via a publicly available API without extensive NLP knowledge or hyperparameter tuning; the final result is a useful model with the ability to extract specialized technical information that is nuanced and semantically complex. Additionally, error correction and normalization may be embedded directly into training examples to reduce the need for post-processing. In essence, one can show an LLM-NERRE model both what it should extract and how it can be condensed and presented.

Like many others, we have found using a human-in-the-loop process can help decrease the time required to collect a full training set[52]. Our particular process is shown in Fig. 2. Annotation for scientific information extraction tasks is often a tedious and error-prone process whereas checking intermediate model outputs for errors is qualitatively easier than creating annotations from scratch. Additionally, fine tuning GPT-3/Llama-2 requires fewer training examples to match or exceed the performance of BERT-based models. Figure 4 shows how performance of the fine-tuned models improves quickly at relatively small training set sizes. However, more and more text-completion pairs are required to achieve the same rate of improvement as training set size is increased.

One limitation of our model is that valid output schema formatting is not rigorously enforced in the generation step. The LLM may, for any given sample, output an unparsable sequence. This is particularly apparent when the inference token limit is less than 512 tokens and the schema is JSON, as JSON schema typically requires a larger number of tokens for correct formatting. For example, a nearly-correct output sequence containing 10+ correct entities may be missing a "}" ending character and therefore will not be parsable. Outputs are nearly always parsable (~99% success rate), especially as the number of training examples increases. Failures predominantly occur when the sample exceeds the prompt-completion token limit of the LLM (early termination), which in this work was 512-1024 tokens for both GPT-3 and Llama-2. Because of this, some abstracts that are too long or too dense with information to be processed with this method. This was the

case in the few unparseable completions where the passage and partial completion exceed the token limit and cut off early before the full completion could be output by the model. This limitation may be mitigated by increasing the token limit up to 2048 (GPT-3) or 4096 (Llama-2); we expect the token limitation will become less of a concern as the maximum token size of such models increase.

Another limitation is the tendency of LLMs to generate or invent information that is not actually present in the input text, a phenomenon termed "hallucination"[53,54] in LLM literature. The main manifestation of hallucination we observed was the addition of names or chemical formulae for a material when only one or the other was mentioned (for example writing "SiO2" in the formula field even though the paragraph only mentions "silica"). Although these hallucinations could potentially be correct, because the source text does not include them, we believe they should not be included in the output of information extraction models. We could enforce this by the requirement that all extracted entities should occur word-for-word in the source text, but the fact that these models do not extract phrases exactly can also be a useful feature because it allows for automatic entity normalization. For example, an abstract may mention both "p-ZnSe doped with N" and "nitrogen-doped ZnSe" in the same passage. Is "doped with N" or "nitrogen-doped" the correct description to extract? Clearly, both are correct and either one could be reasonably chosen. Moreover, "N-doped" could also be extracted and would be factually correct even though "N-doped" never occurs in the passage. Because LLMs can learn implicit normalization rules, if the annotator is consistent in how they normalize cases like this (such as always using "X-doped" and/or "p(n)-type"), the model generally follows the same normalization scheme and it can greatly reduce the amount of entity normalization post-processing required later. We differentiate this from hallucination in that the inference is fully justified by the content in the source text rather than simply plausible.

Finally, the choice of LLM poses a practical tradeoff for researchers: essentially, ease of use vs. control. Using a proprietary LLM such as GPT-3 through an online API enables the LLM in our method to be treated as a "black box", and abstracting away LLM fine-tuning details allows researchers to focus entirely on their domain-specific information extraction tasks. However, the underlying LLM is exclusively controlled by a private entity, posing problems of

reproducibility and security. Regarding security, potentially sensitive or confidential data must be sent to the entity for processing; regarding reproducibility, the models cannot be shared, and the entity controlling the LLM may at any time change the model, amend the fine-tuning method, or revoke access to the model altogether. More, the cost for inference on large datasets using trained models may be prohibitive. In contrast, using self-hosted models such as Llama-2[31] or GPT-NeoX 20B[55] favors control over ease of use. The weights and code for the model are fully accessible, and inference cost is restricted only by the user's budget on a cluster with capable GPUs. However, successfully running, fine-tuning, and deploying LLMs such as Llama-2 on cluster infrastructure is non-trivial for many scientists. Cloud-hosted open-access models (e.g., Llama-2 hosted on a managed cloud instance) may provide a solution to the ease of use vs. control tradeoff, as the technical details of fine-tuning are abstracted away from the user but the fine-tuned models themselves can remain open-access. Similarly, zero-shot approaches without fine-tuning may make scientific information extraction more accessible at the expense of accuracy (see Supplementary Discussion 6). Methods for reducing the number of parameters needed for LLM inference and fine-tuning[56–59] are also a promising avenue for reducing the complexity and cost of self-hosting LLMs. As these methods advance and LLM codebases become more mature, we expect fine-tunable models compatible with LLM-NERRE will become simultaneously powerful, easy to self-host, reproducible, and under researchers' full control. We hope the code examples of both fine-tuning and running inference using the published model weights we provide in Methods are a first step in the direction of powerful and open source NERRE models.

In summary, this work demonstrates that LLMs that are fine-tuned on a few hundred training examples are capable of extracting scientific information from unstructured text and formatting the information in user-defined schemas. This is in contrast to past models which were successful in extracting entities from text but struggled to relate those entities or structure them in meaningful ways. The proposed method is simple and broadly accessible given the APIs and interfaces currently available such as GPT-3. Furthermore, we have made the Llama-2 LoRA weights of all models shown in this paper available for download (see Methods and Code Availability), allowing researchers to investigate the LLM-NERRE method on their own hardware. We expect these advancements to greatly facilitate the rate and accuracy by which historical scientific text can be converted to structured forms.

# Methods

## General sequence-to-sequence NERRE

We fine-tune Llama-2 and GPT-3 models to perform NERRE tasks using 400–650 manually annotated text-extraction (prompt-completion) pairs. Extractions contain the desired information formatted with a predefined, consistent schema across all training examples. These schemas can range in complexity from English sentences with pre-defined sentence structures to lists of JSON objects or nested JSON objects. In principle, many other potential schemas (e.g., YAML, psuedocode) may also be valid, though we do not explore those here. Once fine-tuned on sufficient data adhering to the schema, a model will be capable of performing the same information extraction task on new text data with high accuracy. The model outputs completions in the same schema as the training examples. We refer to this approach generally as "LLM-NERRE".

Our general workflow for training GPT-3 and Llama-2 to perform NERRE tasks is outlined in Fig. 2. Annotations are performed by human domain experts to create an initial training set, and then a partially trained model (GPT-3) is used to accelerate the collection of additional training examples. Fine-tuning is then performed on these examples to produce a "partially trained" model, which is used to pre-fill annotations that are subsequently corrected by the human annotator before being added to the training set. Once a sufficient number of

annotations have been completed, the final fine-tuned model is capable of extracting information in the desired format without human correction. Optionally, as illustrated in Figs. 5 and 6, the structured outputs may be further decoded and post-processed into hierarchical knowledge graphs.

## Task and schema design

**Solid-state impurity doping schema.** The Doping-English and Doping-JSON schemas aim to extract two entity types (host and dopant) and the relations between them (host-dopant), returned as either English sentences or a list of one or more JSON objects. Hosts are defined as the host crystal, sample, or material class along with crucial descriptors in its immediate context (e.g., "ZnO2 nanoparticles", "LiNbO3", "half-Heuslers"). Dopants are taken to be any elements or ions that are minority species, intentionally added impurities, or specific point defects or charge carriers ("hole-doped", "S vacancies"). One host may be doped with more than one dopant (e.g., separate single-doping or co-doping), or the same dopant may be linked to more than one host material. There may also be many independent pairs of dopant-host relations, often within a single sentence, or many unrelated dopants and hosts (no relations). We impose no restriction on the number or structure of the dopant-host relations beyond that each relation connects a host to a dopant. The Doping-JSON schema represents the graph of relationships between hosts and dopants within a single sentence, where unique keys identify dopant and host strings. The model aims to learn this relatively loose schema during fine-tuning. A separate key, "hosts2dopants", describes the pairwise relations according to those unique keys. The Doping-English schema encodes the entity relationships as quasi-natural language summaries. The Doping-English schema represents the same information as the Doping-JSON schema, but more closely mimics the natural language pre-training distribution of the LLMs we tested. When there are multiple items to extract from the same sentence, the output sentences are separated by newlines.

For the DopingExtra-English schema, we introduce two additional entities: modifiers and result, without explicit linking (i.e., NER only). The results entity represents formulae with algebra in the stoichiometric coefficients such as $Al_xGa_{1-x}As$, which are used for experiments with samples from a range of compositions or crystalline solid solutions (e.g., $CaCu_{3-x}Co_xTi_4O_{12}$). We also include stoichiometries where the algebra is substituted (i.e., $x$ value specified) and the doped result is a specific composition (e.g., $CaCu_{2.99}Co_{0.1}Ti_4O_{12}$). Modifiers are loosely bounded entity encapsulating other descriptors of the dopant-host relationship not captured by dopant, host, or result. These can be things like polarities (e.g., "n-type", "n-SnSe"), dopant quantities (e.g., "5 at.%", "$x < 0.3$"), defect types (e.g., "substitutional", "antisite", "vacancy") and other modifiers of the host to dopant relationship (e.g., "high-doping", "degenerately doped"). These entities (host, dopant, result, and modifiers) were chosen to define a minimal effective schema for extracting basic doping information.

All doping-related models are trained to work only on single sentences. The main motivation for this design choice is that the vast majority of dopant-related data can be found within single sentences, and the remaining relational data is often difficult to resolve consistently for both human annotators and models. We expand on problems with annotations and ambiguity in Supplementary Discussion 5 and we further explain the doping task schemas in Supplementary Note 1.

**General materials information schema.** In our previous work[4,5], we focused on NER for a specific set of entity types that are particularly relevant in materials science: materials, applications, structure/phase labels, synthesis methods, etc. However, we did not link these labeled entities together to record their relations beyond a simple "bag-of-

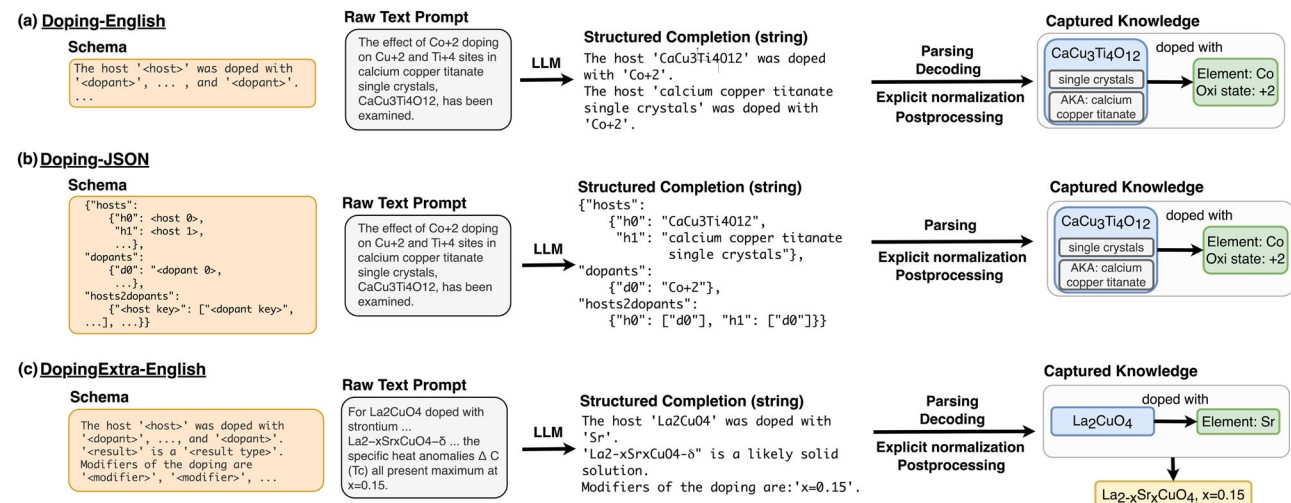

**Fig. 5 | Diagrams of doping information extraction using large language models (LLMs) for joint named entity and relation extraction (NERRE).** In all three panels, an LLM trained to output a particular schema (far left) reads a raw text prompt and outputs a structured completion in that schema. The structured completion can then be parsed, decoded, and formatted to construct relational diagrams (far right). We show an example for each schema (desired output structure). Parsing refers to the reading of the structured output, while decoding refers to the programmatic (rule-based) conversion of that output into JSON form. Normalization and postprocessing are programmatic steps which transform raw strings (e.g., "Co+2") into structured entities with attributes (e.g., Element: Co, Oxidation state +2). **a** Raw sentences are passed to the model with Doping-English schema, which outputs newline-separated structured sentences that contain one `host` and one or more `dopant` entities. **b** Raw sentences are passed to a model with Doping-JSON schema, which outputs a nested JSON object. Each `host` entity has its own key-value pair, as does each `dopant` entity. There is also a list of `host2dopant` relations that links the corresponding dopant keys to each host key. **c** Example for the extraction with a model using the DopingExtra-English schema. This first part of the schema is the same as in **a**, but additional information is contained in `doping modifiers`, and `results`-bearing sentences are included at the end of the schema.

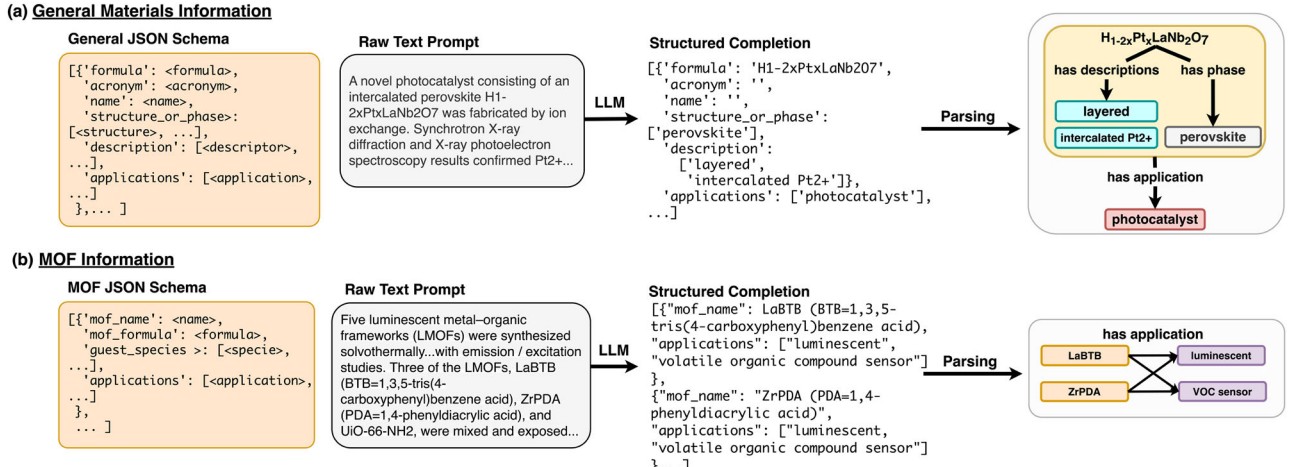

**Fig. 6 | Diagrams of general information extraction and metal organic framework (MOF) information extraction using large language models (LLMs) for joint named entity and relation extraction (NERRE).** In both panels, an LLM trained using a particular schema (desired output structure, far left) is prompted with raw text and produces a structured completion as JSON. This completion can then be parsed to construct relational diagrams (far right). Each task uses a different schema representing the desired output text structure from the LLM. **a** Schema and labeling example for the general materials-chemistry extraction task. Materials science research paper abstracts are passed to an LLM using General-JSON schema, which outputs a list of JSON objects representing individual material entries ordered by appearance in the text. Each material may have a name, formula, acronym, descriptors, applications, and/or crystal structure/phase information. **b** Schema and labeling example for the metal-organic frameworks extraction task. Similar to the General-JSON model, the MOF-JSON model takes in full abstracts from materials science research papers and outputs a list of JSON objects. In the example, only MOF name and application were present in the passage, and both MOFs (LaBTB and ZrPDA) are linked to both applications (luminescent and VOC sensor).

entities" approach. In this work, we train an LLM to perform a "general materials information extraction" task that captures both entities and the complex network of interactions between them.

The schema we have designed for this task encapsulates an important subset of information about solid compounds and their applications. Each entry in the list, a self-contained JSON document, corresponds one-to-one with a material mentioned in the text. Materials entries are ordered by appearance in the text. The root of each entry starts with a compound's name and/or its chemical formula. If a name or formula is not mentioned for a material, no information about that material is extracted from the text. We also extract acronyms mentioned for a material's name/formula, although in cases where only an acronym is mentioned we do not create a material entry for the compound. Compounds that are not solids (ions, liquids, solvents, solutions, etc) are generally not extracted. The `name`, `formula`, and `acronym` fields are exclusively given string value in the JSON document

for each material whereas the `description`, `structure_or_phase`, and `applications` fields are lists of an arbitrary number of strings. We label this model `General-JSON`, and an example is shown in Fig. 6 (a).

`Description` entities are defined as details about a compound's processing history, defects, modifications, or the sample's morphology. For example, consider the hypothetical text "Pt supported on CeO2 nanoparticles infused with Nb...". In this case, the `description` value for the material object referring to "Pt" might be annotated as `"['supported on CeO2']"`, and the `description` entities listed for "CeO2" would be `"['nanoparticles', 'Nb-doped']"`.

`Structure_or_phase` entities are defined as information that directly implies the crystalline structure or symmetry of the compound. Crystal systems such as "cubic" or "tetragonal", structure names such as "rutile" or "NASICON", and space groups such as "Fd3m" or "space group No. 93" are all extracted in this field. We also include any information about crystal unit cells, such as lattice parameters and the angles between lattice vectors. "Amorphous" is also a valid structure/phase label.

`Applications` are defined as high-level use cases or major property classes for the material. For example, a battery cathode material may have `"['Li-ion battery', 'cathode']"` as its applications entry. Generally, applications are mentioned in the order they are presented in the text, except for certain cases such as battery materials, in which case the type of device is generally mentioned before the electrode type, and catalysts, where the reaction catalyzed is generally listed following the "catalyst" entity in the list (e.g.,`"['catalyst', 'hydrogenation of citral']"`).

More details about the general materials information task schema are provided in the Supplementary Discussion 4.

**Metal–organic framework (MOF) schema.** The schema used for the MOF cataloging task is based on the general materials information schema described in the previous section, which was modified to better suit the needs of MOF researchers. We developed this schema to extract MOF names (`name`), an entity for which there is no widely accepted standard[60], and chemical formulae (`formula`), which form the root of the document. If no name or formula is present, no information is extracted for that instance. In addition, because there is a great deal of interest in using MOFs for ion and gas separation[61,62], we extract `guest species`, which are chemical species that have been incorporated, stored, or adsorbed in the MOF. We extract `applications` the MOF is being studied for as a list of strings (e.g., `"['gas-separation']"` or `"['heterogeneous catalyst', 'Diels-Alder reactions']"`) as well as a relevant `description` for the MOF, such as its morphology or processing history, similar to the general information extraction schema. Entries in the list are generally added in the order the material names/formulae appear in the text. The MOF extraction model is labeled `MOF-JSON`, and an example is shown in Fig. 6 (b).

**Comparison baselines and evaluation**
To compare our model with other sequence-to-sequence approaches to information extraction, we perform a benchmark of two methods on the doping task to compare to the LLM-NERRE models. The first employs the seq2rel method of Giorgi et al.[41] for the host-dopant task. We formatted host-dopant relationships under tags labeled @DOPANT@ and @BASEMAT@ (base/host material), with their relationship signified by @DBR@ ("dopant-base material relationship"); these sequences were constructed from the same training data as the `Doping-JSON` and `Doping-English` models. We trained seq2rel to perform sentence-level extraction with 30 epochs, batch size of 4, encoder learning rate $2 \times 10^{-5}$, decoder learning rate $5 \times 10^{-4}$, and pretrained BiomedNLP BERT tokenizer[51] (further training details can be found in the Supplementary Note 4). Additionally, we compare against the previously

published MatBERT doping-NER model[5] combined with proximity-based heuristics for linking (see Supplementary Note 5). With this method, a MatBERT NER model pretrained on ~50 million materials science paragraphs and fine-tuned on 455 separate manually annotated abstracts first extracts hosts and dopants and then links them if they co-occur in the same sentence.

**Datasets**
Datasets were prepared from our database of more than 8 million research paper abstracts[63]. Annotations were performed by human annotators using a graphical user interface built using Jupyter[64], although in principle annotations could be conducted via a simple text editor. To accelerate the collection of training data, new annotations are collected via a "human in the loop" approach where models are trained on small datasets and their outputs are used as starting points and corrected by human annotators (see Fig. 2.) This process of training and annotation is completed multiple times until a sufficiently large set of training data was achieved. Each dataset was annotated by a single domain expert annotator. Class support for each annotated dataset is provided in Supplementary Tables 1-3.

**Doping dataset.** Training and evaluation data was gathered from our database of research paper abstracts using the keywords "n-type", "p-type", "-dop", "-codop", "doped", "doping", and "dopant" (with exclusions for common irrelevant keywords such as "-dopamine"), resulting in ~375k total abstracts. All doping tasks were trained on text from 162 randomly selected abstracts, comprising 1215 total sentences and filtered with regular expressions to only include 413 relevant (potentially including doping information) sentences. Doping tasks were tested on an additional 232 sentences (77 relevant by regex) from a separate holdout test set of 31 abstracts.

**General materials dataset.** Training and evaluation data was gathered from our abstract database by using keywords for a variety of materials properties and applications (e.g., "magnetic", "laser", "space group", "ceramic", "fuel cell", "electrolytic", etc). For each keyword a materials science domain expert annotated ~10–50 abstracts, which resulted in ≈650 entries manually annotated according to the general materials information schema. Results were evaluated using a 10% random sample for validation, and this procedure was averaged over five trials using different random train/validation splits with no hyperparameter tuning.

**Metal–organic framework dataset.** Training and evaluation data was selected from our database using the keywords "MOF", "MOFs", "metal-organic framework", "metal organic framework", "ZIF", "ZIFs", "porous coordination polymer", and "framework material", which produced approximately 6,000 results likely containing MOF-related information. From these, 507 abstracts were randomly selected and annotated by a MOF domain expert. Results were evaluated using the same repeated random split procedure as the general materials dataset in the previous section.

**GPT-3 fine tuning details**
For all tasks, we fine-tune GPT-3 ('davinci', 175B parameters)[26] using the OpenAI API, which optimizes the cross-entropy loss on predicted tokens. Doping models were trained for 7 epochs at a batch size of 1, with inference temperature of 0 and output limited to a maximum length of 512 tokens (all doping models) or 1024 tokens (General-JSON, MOF-JSON). The intermediate models shown in Fig. 4 were trained with a number of epochs depending on the number of training samples $t$: 2 epochs for $2^0 \leq t < 2^6$, 4 epochs for $2^6 < t \leq 2^7$, and 7 epochs for $t \geq 2^8$. Models for the MOF and general materials extraction tasks were trained for 4 epochs with a batch size of 1. We use a learning rate multiplier of 0.1 and a prompt loss weight of 0.01 but have not performed hyperparameter tuning for these hyperparameters. For all

tasks, the start and end tokens used were "`\n\n\n###\n\n\n`" and "`\n\n\nEND\n\n\n`".

## Llama-2 fine-tuning details

Llama-2[31] fine-tunes were performed using a modified version of the Meta Research Llama-2 recipes repository; the modified repository can be found at https://github.com/lbnlp/nerre-llama. Llama-2 fine-tunes were performed using the 70 billion parameter version of Llama-2 (`llama-2-70b-hf`) with quantization (8 bit precision). The number of epochs was set to 7 for doping tasks and 4 for the MOF/general tasks. Llama-2 fine-tunes used parameter efficient fine-tuning (PEFT) using low rank adaptation (LoRA)[58] with LoRA $r = 8$, $\alpha = 32$ and LoRA dropout of 0.05. Further hyperparameter tuning was not performed. Decoding was done without sampling using greedy decoding to be consistent with GPT-3 decoding setting of temperature = 0, with max tokens = 512 for doping task and 1024 for general and MOF task. More details on the fine tuning and inference parameters are available in the modified repository and Supplementary Note 3. All fine-tuning and inference was performed on a single A100 (Ampere) tensor core GPU with 80GB VRAM.

The fine-tuned weights for each model are provided in the NERRE-Llama repository (url above) along with code and instructions for downloading the weights, instantiating the models, and running inference.

## Evaluation criteria

The fuzzy and complex nature of the entities and relationships detailed in the previous section necessitates the use of several metrics for scoring. We evaluate the performance of all models on two levels:

1. A relation $F_1$ computed on a stringent exact word-match basis (i.e., how many words are correctly linked together exactly as they appear in the source text prompt).
2. A holistic information extraction $F_1$ based on manual inspection by a domain expert, which doesn't require words to match exactly.

We separately provide a sequence-level error analysis in Supplementary Note 7 and Supplementary Discussion 1.

**NERRE performance.** We measure NERRE performance as the ability of the model to jointly recognize entities and the relationships between them.

### Exact word-match basis scoring

We score named entity relationships on a word-basis by first converting an entity $E$ into a set of constituent $k$ whitespace-separated words $E = \{w_1, w_2, w_3, ..., w_k\}$. When comparing two entities $E^{\text{true}}$ and $E^{\text{test}}$ that do not contain chemical formulae, we count the number of exactly matching words in both sets as true positives ($E^{\text{true}} \cap E^{\text{test}}$) and the mathematical set differences between the sets as false positives ($E^{\text{test}} - E^{\text{true}}$) or false negatives ($E^{\text{true}} - E^{\text{test}}$). For example, if the true entity is "Bi2Te3 thin film" and the predicted entity is "Bi2Te3 film sample", we record two true positive word exact matches ("Bi2Te3", "film"), one false negative ("thin"), and one false positive ("sample"). Formula-type entities are crucial for identifying materials, so in cases where entities contain chemical formulae, $E^{\text{test}}$ must contain all $w_i$ that can be recognized as stoichiometries for any of $w_i \in E^{\text{test}}$ to be considered correct. For example, if the true entity is "Bi2Te3 thin film", and the predicted entity is "thin film", we record three false negatives. Thus, any formula-type entity (Doping `host`, Doping `dopant`, General `formula`, and MOF `mof_formula`) containing a chemical composition is entirely incorrect if the composition is not an exact match. This choice of evaluation was made to avoid metrics measuring the performance of the model in a misleading way. For example, "Bi2Te3 nanoparticles" and "Bi2Se3 nanoparticles" have very high similarities via Jaro-Winkler (0.977) and character-level BLEU-4 (0.858), but these

two phrases mean entirely different things—the material's chemistry is wrong. Under our scoring system, they are recorded as entirely incorrect because the compositions do not match.

We score relationships between entities on a word-by-word basis to determine the number of correct relation triplets. Triplets are 3-tuples relating word $w_j^n$ of an entity $E_n$ to word $w_k^m$ of an entity $E_m$ by relationship $r$, represented as $(w_j^n, w_k^m, r)$. The total set of correct relationships $T^{\text{true}}$ for a text contains many of these triplets. A test set of relationships $T^{\text{test}}$ is evaluated by computing the number of triplets found in both sets ($T^{\text{true}} \cap T^{\text{test}}$) as true positives and the differences between these sets as false positives ($T^{\text{test}} - T^{\text{true}}$) or false negatives ($T^{\text{true}} - T^{\text{test}}$). Entity triplets are also bound to the same requirement for composition correctness if either of the words in the triplet belong to an formula-type entity (`host`, `dopant`, `formula`, `mof_formula`), i.e., we count all triplets for two entities as incorrect if the formula is not an exact string match. With correct and incorrect triplets identified, $F_1$ scores for each relation are calculated as:

$$\text{precision} = \frac{\text{No. of correct relations retrieved}}{\text{No. of relations retrieved}} \quad (1)$$

$$\text{recall} = \frac{\text{No. of correct relations retrieved}}{\text{No. of relations in test set}} \quad (2)$$

$$F_1 = \frac{2(\text{precision} \cdot \text{recall})}{\text{precision} + \text{recall}} \quad (3)$$

To compute triplet scores across entire test sets in practice, we first select a subset of relations to evaluate. We note that this is not a full evaluation of the task we are training the model to perform, which involves linking many interrelated entities simultaneously, but is rather provided to help give a general sense of its performance compared to other NERRE methods. For the doping task, we evaluate `host-dopant` relationships. For the general materials and MOF tasks, we evaluate relationships between the formula field (`formula` for general materials, `mof_formula` for MOFs) and all other remaining fields. For `description`, `structure_or_phase`, and `applications` fields, all of which may contain multiple values, all of the possible formula-value pairs are evaluated.

### Manual evaluation

The metrics provided in prior sections demonstrate automatic and relatively strict methods for scoring NERRE tasks, but the underlying capabilities of the LLM models are best shown with manual evaluation. This is most apparent in the case of the `General-JSON` model, where exact boundaries on entities are fuzzier, precise definitions are difficult to universally define, and annotations include some implicit entity normalization. For example, the text "Pd ions were intercalated into mesoporous silica" may have equivalently have a correct `description` field for the material "silica" including "Pd-intercalated", "Pd ion-intercalated", "intercalated with Pd ions", etc.; the exact choice of which particular string is used as the "correct" answer is arbitrary.

To better address scoring of these fuzzy tasks, we introduce an adjusted score based on a domain expert's manual evaluation of whether the information extracted is a valid representation of the information actually contained in the passage. We term this adjusted score "manual score"; it constitutes a basis for precision, recall, and $F_1$ that quantifies the quality of overall information capture for cases where there may be equivalent or multiple ways of representing the same concept. This score was constructed to better estimate the performance of our model for practical materials information extraction tasks.

We score entities extracted by annotators but not present in the model's output as false negatives, except when reasonable variations are present. The criteria for a true positive are as follows:

1. The entity comes from the original passage or is a reasonable variation of the entity in the passage (e.g., "silicon" ⟶ "Si"). It is not invented by the model.
2. The entity is a root entity or is grouped with a valid root entity. For the `General-JSON` model, a root entity is either a material's formula or name. If both are present, the formula is used at the root.
3. The entity is in the correct field in the correct root entity's group (JSON object).

Manual scores are reported per-entity as if they were NER scores. However, the requirements for a true positive implicitly include relational information, since an entity is only correct if is grouped with the correct root entity.

### Reporting summary

Further information on research design is available in the Nature Portfolio Reporting Summary linked to this article.

## Data availability

All data used for this study are available at https://github.com/LBNLP/NERREand via Zenodo[65], which contains the annotated datasets and test and train splits. Intermediate files for each step of the pipeline reported in this method are stored in this repository with corresponding documentation. Data for running Llama-2 models are available in the supplementary repository https://github.com/lbnlp/nerre-llama[66]; LoRA weights for all Llama-2 models reported in this paper can be downloaded directly from Figshare (https://doi.org/10.6084/m9.figshare.24501331.v1)[67]. Source data are provided with this paper.

## Code availability

The code used for this study is available at https://github.com/LBNLP/NERREand via Zenodo[65] alongside the data. This code includes Jupyter notebooks for annotation as well as Python scripts for annotation, preprocessing, model training, and model evaluation on the train and test sets presented in this publication. The supplementary repository https://github.com/lbnlp/nerre-llama[66] contains code and data for fine-tuning and inference with Llama-2 models trained in this study, including access to the complete sets of weights via a script.

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

## Acknowledgements

This work was supported by Toyota Research Institute through the Accelerated Materials Design and Discovery program. A.S.R. acknowledges support via a Miller Research Fellowship from the Miller Institute for Basic Research in Science, University of California, Berkeley. Funding for training and evaluating the Llama-2 model was provided by the U.S. Department of Energy, Office of Science, Office of Basic Energy Sciences, Materials Sciences and Engineering Division under Contract No. DE-AC02-05CH11231 (D2S2 program KCD2S2). This research used resources of the National Energy Research Scientific Computing Center (NERSC), a U.S. Department of Energy Office of Science User Facility located at Lawrence Berkeley National Laboratory, operated under Contract No. DE-AC02-05CH11231 using NERSC award BES-ERCAP0024004. We thank Anna Sackmann (Science Data and Engineering Librarian at UC Berkeley) for helping us to obtain Text and Data Mining agreements with the specified publishers and we also thank J. Montoya and A. Trewartha for helpful discussions.

## Author contributions

J.D., A.D., N.W., and A.J. developed the information extraction method presented in this paper and J.D. and A.D. collected the abstract dataset used. J.D originated and performed supporting experiments to justify the sequence-to-sequence approach and use of GPT-3 for document-level information extraction from materials science text. A.D. expanded on JD's initial experiments to further develop the approach. N.W. and J.D. developed the sequence-to-JSON method. A.D. created the doping schemas, developed the sequence-to-sentence method for doping, and annotated the doping dataset. J.D. created the general materials information schema, annotated the general materials dataset, trained the general materials information extraction model, and manually scored the information extraction results for the General-JSON model validation set results. A.D. trained all Doping-* LLM-NERRE models and implemented the MatBERT + Proximity doping model. S.L. trained and collected data from the Llama-2 and seq2rel models. A.D. performed the learning curve experiments, and collected data on annotation times. A.S.R. and J.D. co-created the MOF schema and annotated the MOF dataset while J.D. trained the MOF-JSON model. In addition to intellectual leadership and overall research directions, A.J. also contributed to task design and task scoring metrics. G.C., K.P., and A.J. supervised this work. All authors contributed to writing the manuscript.

## Competing interests

The authors declare no competing interests.
