## [Peer Review File · Nature Communications]

REVIEWER COMMENTS

Reviewer #1 (Remarks to the Author):

The manuscript “Structured information extraction from scientific text with large language models” by Dagdelen et al. is a comprehensive study to extract complex information (compound names, symbols, representations, applications, and properties) from scientific text using GPT-3 models. The manuscript implements a novelty and complex task that uses a sequence-to-sequence approach to document-level joint named entity recognition and relation extraction (NERRE) for the extraction of complex information from scientific text. The work is significant to the materials field and may be applied to other chemistry-related subjects. The work supports the conclusions and claims, and the methodology is sound.

I could not find any flaws in the data analysis, interpretation, and conclusions because the work does not meet the expected standards in the computational science field and there is not enough detail provided for the work to be reproduced. These prohibit publication or require revision. The authors MUST be transparent about the availability of the data, clarity of the methodologies and software, and reproducibility of the presented work. Therefore, authors are required to provide information about the data and software described and used in their manuscript in a separate Data and Software Availability section at the end of the manuscript. The authors MUST include one or several tutorial files (*.ipynb) to explain the work in a better fashion, allowing the referee and readers to check the reproducibility.

Major concern:

Comment 1: the github www.github.com/LBNLP/NERRE does not exist. And, if the github www.github.com/LBNLP/ is searched, the repository on named-entity recognition (NER, MatBERT-NER) is found, but it does not contain the annotated datasets, test and train splits, evaluation code, and jupyter notebooks containing the annotation UI mentioned in the data availability section. So, it is impossible to check or reproduce the manuscript, which is not reasonable.

It is very important that the authors include a tutorial as jupyter notebooks.

Minor concerns:

Comment 1: The authors missed a literature reference (10.3389/fchem.2022.930369) in the very beginning of introduction.

Comment 2: Please, remove the word “also” from “these models may also be also adept at complex scientific information extraction”.

Comment 3: Please, correct “Figs. 5–??” to “Figs. 5–6”.

Comment 4: There is a 'Nb-doped']" lost in the middle of text.

Reviewer #2 (Remarks to the Author):

The authors present an approach to joint named entity recognition and relation extraction in three tasks in materials chemistry using a fine-tuned large language model (GPT3). The approach is a novel approach to trivially extract very specific and specialized scientific information from research papers. The human-in-the-loop annotation experiment, in particular, seems to be a powerful demonstration of the potential of LLMs.

I am not an expert in materials chemistry therefore I cannot comment on the utility for experts in that field however the fine-tuning approach and general methodology are all sound. Indeed, the authors provide sufficient detail and demonstration for the approach to be replicated in other scientific domains therefore I think this work is appropriate and relevant for publication in Nature Communications.

However, I think the work is merely skimming the top of the barrel in terms of LLM ability and the results are likely a small snapshot of what LLMs are capable of in regard to LLMs assisting scientific research. An important limitation is that GPT3 is fairly old (by machine learning research standards) and presents some issues for replication and extension of this work because the model is not open-source. In the future, I encourage the authors to repeat their experiments on more recent open-source language models (LLaMA, GPTJ & Neo).

Reviewer #3 (Remarks to the Author):

In the article the authors present an innovative approach for parsing scientific literature using fine-tuned large language models. This study offers a compelling solution to a complex problem and demonstrates its application in the field of materials science. By utilizing the GPT-3 fine-tuning API, the authors effectively train models to extract information about materials from unstructured text. Their analysis of model performance, comparison of different schemas, and evaluation of a human-in-the-loop approach are particularly interesting.

However, there are several revisions that could improve the manuscript:

1. The authors should clarify the performance of GPT-3/3.5/4 in few-shot settings. They could select a few of the most challenging examples from the dataset and use them in the prompt with a textual description of the problem.
2. It would be interesting to examine the zero-shot performance in the same setting, using a prompt such as, "extract formulas, descriptions, and applications for every material appearing in the text and present it as a JSON file."
3. The contents of Tables II and III are somewhat confusing. Table III appears to contain information about entity recognition scores (manual), while Table II focuses on relation extraction (automatic). When comparing the values in the text, the authors use the General-JSON section of Table II. However, it remains unclear what the 0.613 value for the formula score refers to. Additionally, some values (e.g., recall for names) are lower in Table III than in Table II, though the automatic evaluation should provide a lower bound for the scores.
4. As the authors mention, open-source language models might be able to reproduce the results demonstrated in this study. Including experiments with such models would be a valuable addition.

In summary, this paper offers a good contribution to the field and could be published in Nature Communications after addressing the suggested revisions.

Responses to Reviewer Comments

General Response:

We thank all reviewers for their careful reading and constructive comments.

Manuscript: To address the reviewers' and editors' comments about reproducibility, we have rerun the main NERRE results using fine-tuned Llama-2+LoRA models including all applicable entity relationships and NER scores. Our results with Llama-2 70 billion parameter models, using identical NERRE procedure to our previous manuscript, are similar to the results obtained with GPT-3; these results suggest our method is robust to the choice of LLM. The fine-tuned model weights and code for all Llama-2 models are available in a repository (<https://github.com/lbnlp/nerre-llama>), and the code for annotating and scoring models is in our original repository (<https://github.com/lbnlp/NERRE>). We provide details on all changes below.

While fine-tuning and evaluating the Llama-2 models, we identified two errors in the results originally submitted. The errors have been corrected and resulted in slight changes to the reported scores, but without affecting the overall conclusions of the manuscript.

- **Errors in scoring code:** We identified an issue in the exact word-match scoring code affecting all NERRE results causing certain entity-links to be double counted. After correcting this error, the NERRE linking scores across all models (including baselines) dropped by ~0.1 F1-score with the General task most affected. The manual scores (previously called "manual information extraction scores") are unaffected. We have updated the code and data in the repository to reflect the fixed scoring script.
- **Errors in annotations:** We have made corrections to 52 erroneous annotations of the 507 total annotations of the MOF-JSON task. These erroneous annotations were caused by a software issue in the annotation UI, and they resulted in lower NERRE scores of the MOF-JSON models in some results. We have updated the data in the repository, retrained the MOF-JSON models (both GPT-3 and Llama-2) using the correct data, and used these models to update the MOF NERRE scores in the main text and supplement. Aside from using the correct annotations for the MOF task, the methodology remains identical to that shown in the original manuscript. Including the correct annotations in the MOF dataset significantly increased the overall NERRE performance for MOF-JSON. Tables S1-S2 of the supplementary information have also been updated with results from the corrected models. The corrected models broadly have slightly higher parsability, Jaro-Winkler similarity, exact match sequence accuracy, exact match triplet relationships, and NER scores across most MOF entities.

To reflect the expanded scope of the data presented, including the new tests of Llama-2+LoRA models, we have separated Table II of the original manuscript into two separate tables, the new Tables II and IV. Table II of the original manuscript was:

Method (Input Type)	Relation	Precision (Exact Match)	Recall (Exact Match)	F1 (Exact Match)
MatBERT-Proximity (sentence)	host-dopant	0.441	0.714	0.545
Seq2rel (sentence)	host-dopant	0.550	0.650	0.600
Doping-English (sentence)	host-dopant	0.789	0.776	0.783
Doping-JSON (sentence)	host-dopant	0.758	0.684	0.719
DopingExtra-English (sentence)	host-dopant	0.872	0.828	0.849
General-JSON (paragraph)	formula-name	0.716	0.539	0.613
	formula-acronym	0.635	0.470	0.537
	formula-application	0.499	0.470	0.481
	formula-structure/phase	0.411	0.368	0.388
	formula-description	0.392	0.304	0.341
MOF-JSON (paragraph)	name-formula	0.455	0.409	0.424
	name-application	0.560	0.461	0.504
	name-guest specie	0.665	0.588	0.606
	name-description	0.464	0.247	0.318

The revised manuscript now re-organizes the updated data as follows, with one table for all LLM-NERRE models using a JSON schema across all tasks (doping, MOFs, general), and one table using the doping task as an ablation study on the effect of schema (English sentences vs. JSON) on NERRE performance.

- Table II: Main NERRE results using JSON schema for all models:

TABLE II: Named entity recognition and relation extraction scores for models using a JSON output schema. Exact match scores are evaluated on a per-word basis, and links are only correct if both entities and the relationship are correct. The exact match metric scores output that contains the correct information but is written differently as incorrect, making such scores a lower bound on the true performance of models. F_1 , precision, and recall are reflect the scores on a hold out test set for doping models and averages over five cross-validation sets for the general and MOF models. Best F_1 scores for each task are shown in bold.

Task	Relation	GPT-3			LLaMA2		
		E.M. Precision	E.M. Recall	E.M. F_1	E.M. Precision	E.M. Recall	E.M. F_1
Doping	host-dopant	0.772	0.684	0.726	0.836	0.807	0.821
General	formula-name	0.507	0.429	0.456	0.462	0.417	0.367
	formula-acronym	0.500	0.250	0.333	0.333	0.250	0.286
	formula-structure/phase	0.538	0.439	0.482	0.551	0.432	0.47
	formula-application	0.542	0.543	0.537	0.545	0.496	0.516
	formula-description	0.362	0.35	0.354	0.347	0.342	0.340
MOF	name-formula	0.425	0.688	0.483	0.460	0.454	0.276
	name-guest specie	0.789	0.576	0.616	0.497	0.407	0.408
	name-application	0.657	0.518	0.573	0.507	0.562	0.531
	name-description	0.493	0.475	0.404	0.432	0.411	0.389

- Table III: Manual scores (unchanged)
- Table IV: Effect of schema on NERRE scores for the doping task, and comparison to baseline models (seq2rel, MatBERT-Proximity).

TABLE IV: Comparison of LLMs with different NERRE schemas to baseline models on host-dopant extraction task. NERRE exact match scores are evaluated on a per-word basis, and links are only correct if both entities and the relationship are correct. DopingExtra-English scores here refer to only host-dopant relation prediction. We note that exact match scores output that contains the correct information but is written differently as incorrect, making such scores a lower bound on the true performance of models. F_1 , precision, and recall are computed on a hold-out test set from 77 sentences. Best scores for precision, recall, and F_1 are shown in bold.

Model	Schema	Precision (Exact Match)	Recall (Exact Match)	F_1 (Exact Match)
MatBERT-Proximity	n/a	0.377	0.403	0.390
Seq2rel	n/a	0.420	0.605	0.496
GPT-3	Doping-JSON	0.772	0.684	0.725
	Doping-English	0.803	0.754	0.778
	DopingExtra-English	0.820	0.798	0.809
LLaMA2	Doping-JSON	0.836	0.807	0.821
	Doping-English	0.787	0.842	0.814
	DopingExtra-English	0.694	0.815	0.750

In summary, we found that the results of the Llama-2 models suggest that similar results can be obtained through publicly available models as through the API-based GPT-3. We found GPT-3 slightly outperformed Llama-2 on the General and MOF tasks, but Llama-2 was the preferred model for the doping task. Within the doping task, varying only the schema, GPT-3 models generally performed better with the natural language-like Doping-English/DopingExtra-English schemas, while Llama-2 performed best using the Doping-JSON schema. We have also updated the supplementary information with more granular information about the support of each class, the NER scores for each model per-entity, and details on the model fine-tuning and inference.

To fully address the issues in scoring mentioned previously, we also re-scored the learning curve figure (Figure 4) examining NERRE performance as a function of fine-tuning set size. The original figure 4 was:

FIG. 4: Test set performance by number of training samples for the doping extraction task using the Doping-English model, which specifically requires the model to learn a new and specific sentence structure to use as the output. We note that below approximately 10 samples, the scores are zero because the model has not learned the specific structure of the desired output sentences.

The updated Figure 4 is:

FIG. 4: Test set performance by number of training samples for the doping extraction task using the Doping-English model, which specifically requires the model to learn a new and specific sentence structure to use as the output. We note that below approximately 10 samples, the scores are zero because the model has not learned the specific structure of the desired output sentences.

We have also included a brief methods section on fine-tuning Llama-2 with the full parameter set in the supplementary information. Additionally, we have updated the results and discussion text to briefly discuss our findings on GPT-3 vs. Llama-2 and the limitations of both. However, the core conclusions of the paper are unchanged. We believe the similar performance of our method using Llama-2 supports the claim this method may also work on LLMs outside of GPT-3.

Minor changes: We have clarified the terminology of “model” vs. “schema” throughout the main text. We have also added references to relevant work by Zheng et al (<https://doi.org/10.1021/jacs.3c05819>), Xie et al (<https://arxiv.org/abs/2308.13565>), and Jablonka et al (<https://doi.org/10.26434/817chemrxiv-2023-fw8n4-v2>, <https://arxiv.org/abs/2306.06283>). We have added a reference to Oliveira et al (<https://doi.org/10.3389/fchem.2022.930369>) at the recommendation of Reviewer 2, along with references to alternative GPT models. We have clarified which results come from Llama-2 (Tables II, IV) and which results come from GPT-3 (Table III, learning curve/Fig 4, annotation time graph/Fig 3). Finally, we have corrected a few typos, grammatical errors, and clarity issues.

Code and Data: We have made all data and code for available in the repository <https://github.com/lbnlp/NERRE> with instructions and documentation. This data and code is also available in the zip file attached to this submission. We have also included scripts and jupyter notebooks for annotation, where applicable. We have also included supplementary code for fine-tuning Llama-2 in the repository <https://github.com/lbnlp/nerre-llama> along with scripts to download the weights so readers can easily load and run the Llama-2 models reported in the manuscript.

Reviewer #1 (Remarks to the Author):

The manuscript “Structured information extraction from scientific text with large language models” by Dagdelen et al. is a comprehensive study to extract complex information (compound names, symbols, representations, applications, and properties) from scientific text using GPT-3 models. The manuscript implements a novelty and complex task that uses a sequence-to-sequence approach to document-level joint named entity recognition and relation extraction (NERRE) for the extraction of complex information from scientific text. The work is significant to the materials field and may be applied to other chemistry-related subjects. The work supports the conclusions and claims, and the methodology is sound. I could not find any flaws in the data analysis, interpretation, and conclusions because the work does not meet the expected standards in the computational science field and there is not enough detail provided for the work to be reproduced. These prohibit publication or require revision. The authors MUST be transparent about the availability of the data, clarity of the methodologies and software, and reproducibility of the presented work. Therefore, authors are required to provide information about the data and software described and used in their manuscript in a separate Data and Software Availability section at the end of the manuscript. The authors MUST include one or several tutorial files (.ipynb) to explain the work in a better fashion, allowing the referee and readers to check the reproducibility.*

Major concern:

Comment 1: the github www.github.com/LBNLP/NERRE does not exist. And, if the github www.github.com/LBNLP/ is searched, the repository on named-entity recognition (NER, MatBERT-NER) is found, but it does not contain the annotated datasets, test and train splits, evaluation code, and jupyter notebooks containing the annotation UI mentioned in the data availability section. So, it is impossible to check or reproduce the manuscript, which is not reasonable.

It is very important that the authors include a tutorial as jupyter notebooks.

Response: We thank the reviewer for noticing this, as we had not made the repository public at the time of submission. We apologize for the oversight. It is now public with updated documentation and the code the reviewer has requested. The repository link here should work: <https://github.com/lbnlp/NERRE>. The code is segmented into two sections: one section for the more complex MOF and General extraction tasks (all using JSON schema), and one section for all Doping tasks with all schemas. This separation was made to keep the doping code separate, since we use this code to assess the effect of schema choice on performance - this requires converting strings (e.g., English sentences) to/from relational JSON data for the three schemas we present in the manuscript. We include all intermediate files (including train/test splits) used as input or output with the models used in the manuscript (although the GPT-3 models require access to our private API key) so that each step can be independently reproduced by interested parties.

We include the original annotation CLI script used to annotate the doping dataset as well as a Jupyter notebook UI for annotating the General/MOF tasks. The core code for training, predicting, and evaluating the models is given as CLI-compatible python and shell scripts; the scripts and their software requirements are documented in step-by-step manner so interested parties can create their own models with exactly the same method as presented in the manuscript.

We have also included code for running (and reproducing) our fine-tuned models with Llama-2 in this separate repository (<https://github.com/lbnlp/nerre-llama>). The properly formatted output files from this repository should work with the scoring code in the main repository to obtain the same scores we report in the manuscript.

Minor concerns:

Comment 1: The authors missed a literature reference (10.3389/fchem.2022.930369) in the very beginning of introduction.

Response: We have included this citation. The text now reads: “Moreover, machine learning models for direct property prediction are being increasingly employed as screening steps for materials discovery and design workflows¹⁻³, but this approach is limited by the amount of

training data available in tabulated databases.” with the new citation “3. Oliveira, O. N. & Oliveira, M. C. F.

Materials discovery with machine learning and knowledge discovery. *Frontiers in Chemistry* 10 (2022). URL <https://doi.org/10.3389/fchem.2022.930369>”

Comment 2: Please, remove the word “also” from “these models may also be also adept at complex scientific information extraction”.

Response: We have fixed this grammatical error. The text now reads “It stands to reason that these models may also be adept at complex scientific information extraction.”

Comment 3: Please, correct “Figs. 5-??” to “Figs. 5-6”.

Response: We have fixed this formatting error. The text now reads “Optionally, as illustrated in Figs. 5 and 6, the structured outputs may be further decoded and post-processed into hierarchical knowledge graphs.”

Comment 4: There is a ‘Nb-doped’]” lost in the middle of text.

Response: The ‘Nb-doped]’ is a continuation of the previous line (on the prior page) and is an artifact from the manuscript formatting. The entire sentence reads:

“In this case, the description value for the material object referring to “Pt” might be annotated as “[‘supported on CeO2]”, and the description entities listed for “CeO2” would be “[‘nanoparticles’, ‘Nb-doped]’.”

Reviewer #2 (Remarks to the Author):

The authors present an approach to joint named entity recognition and relation extraction in three tasks in materials chemistry using a fine-tuned large language model (GPT3). The approach is a novel approach to trivially extract very specific and specialized scientific information from research papers. The human-in-the-loop annotation experiment, in particular, seems to be a powerful demonstration of the potential of LLMs.

I am not an expert in materials chemistry therefore I cannot comment on the utility for experts in that field however the fine-tuning approach and general methodology are all sound. Indeed, the authors provide sufficient detail and demonstration for the approach to be replicated in other scientific domains therefore I think this work is appropriate and relevant for publication in Nature Communications.

However, I think the work is merely skimming the top of the barrel in terms of LLM ability and the results are likely a small snapshot of what LLMs are capable of in regard to LLMs assisting scientific research. An important limitation is that GPT3 is fairly old (by machine learning research standards) and presents some issues for replication and extension of this work because the model is not open-source. In the future, I encourage the authors to repeat their

experiments on more recent open-source language models (LLaMA, GPTJ & Neo).

Response: We thank the reviewer for their comments.

However, I think the work is merely skimming the top of the barrel in terms of LLM ability and the results are likely a small snapshot of what LLMs are capable of in regard to LLMs assisting scientific research:

Response: We agree that our method does not exhaust LLMs' ability to assist scientific research; there may be many paradigms of where complex seq2seq models are able to advance science (e.g., question-answering or use of embeddings for supervised learning models). The ability of these models to seemingly reason (e.g., write code, solve mathematical problems, follow complex chains of logic) implies they could be used for more complex scientific tasks. For example, also consider Jablonka et al.'s (<https://doi.org/10.26434/chemrxiv-2023-fw8n4-v2>) early use of GPT for performing regression and classification experiments or Jablonka et al.'s (<https://arxiv.org/abs/2306.06283>) survey of GPT in a variety of chemistry related tasks.

We have added a sentence to the introduction citing these works; the text now reads:

“The method is able to flexibly handle complex inter-relations (including cases where information exists as lists of multiple items) without requiring enumeration of all of possible n-tuple relations or preliminary NER. Our approach differs from the supervised learning (e.g., regression and classification for chemistry) and inverse design approaches of Jablonka et al.^{47,48} and Xie et al.⁴⁶; rather than using LLMs to directly influence design or predict properties, we aim to (accurately) extract structured hierarchies of information for use with downstream models. We fine-tune a pretrained large language model to accept a text passage (for example, a research paper abstract)...”

With the new citations:

46. Xie, T., Wan, Y., Huang, W., Yin, Z., Liu, Y., Wang, S., Linghu, Q., Kit, C., Grazian, C., Zhang, W., Razzak, I. Hoex, B. DARWIN Series: Domain Specific Large Language Models for Natural Science (2023) URL <https://arxiv.org/abs/2308.13565>.

47. Jablonka, K. M., Schwaller, P., Ortega-Guerrero, A. & Smit, B. Is GPT all you need for low-data discovery in chemistry? (2023). URL <https://doi.org/10.26434/817chemrxiv-2023-fw8n4-v2>.

48. Jablonka, K. M. et al. 14 examples of how llms can transform materials science and chemistry: A reflection on a large language model hackathon (2023). arXiv:2306.06283.

An important limitation is that GPT3 is fairly old (by machine learning research standards) and presents some issues for replication and extension of this work because the model is not open-source. In the future, I encourage the authors to repeat their experiments on more recent open-source language models (LLaMA, GPTJ & Neo):

We heartily agree with the reviewer that this work represents just the beginning of the value that large language models will unlock for the scientific community and thank them for their suggestions, which are a promising line of next steps for our research. **We have repeated the core experiments in the manuscript using the 70 billion parameter version of Llama-2 with LoRA and updated the methods and results accordingly.** In addition to the updates to the results in the earlier part of this response, we have amended the end of the discussion section discussing GPT-3's limitations to now read:

“Finally, the choice of LLM poses a practical tradeoff for researchers: essentially, ease of use vs. control. Using a proprietary LLM such as GPT-3 through an online API enables the LLM in our method to be treated as a “black box”, and abstracting away LLM fine-tuning details allows researchers to focus entirely on their domain-specific information extraction tasks. However, the underlying LLM is exclusively controlled by a private entity, posing problems of reproducibility and security. Regarding security, potentially sensitive or confidential data must be sent to the entity for processing; regarding reproducibility, the models cannot be shared, and the entity controlling the LLM may at any time change the model, amend the fine-tuning method, or revoke access to the model altogether. More, the cost for inference on large datasets using trained models may be prohibitive. In contrast, using self-hosted models such as Llama-2¹⁴ or GPT-NeoX 20B⁵³ favors control over ease of use. The weights and code for the model are fully accessible, and inference cost is restricted only by the user’s budget on a cluster with capable GPUs. However, successfully running, fine-tuning, and deploying LLMs such as Llama-2 on cluster infrastructure is non-trivial for many scientists. Cloud-hosted open-access models (e.g., Llama-2 hosted on a managed cloud instance) may provide a solution to the ease of use vs. control tradeoff, as the technical details of fine-tuning are abstracted away from the user but the fine-tuned models themselves can remain open-access. Furthermore, methods for reducing the number of parameters needed for LLM inference and fine-tuning⁵⁴⁻⁵⁷ are a promising avenue for reducing the complexity and cost of self-hosting LLMs. As these methods advance and LLM codebases become more mature, we expect fine-tunable models compatible with LLM-NERRE will become simultaneously powerful, easy to self-host, reproducible, and under researchers’ full control. We hope the provided code examples of both fine-tuning and running inference using the published model weights we provide in Methods are a first step in the direction of powerful and open source NERRE models.”

Including the new citations:

53. Black, S. et al. Gpt-neox-20b: An open-source autoregressive language model (2022). arXiv:2204.06745.
54. Frantar, E. & Alistarh, D. Sparsegpt: Massive language models can be accurately pruned in one-shot (2023). URL <https://arxiv.org/abs/2301.00774>
55. Sun, M., Liu, Z., Bair, A. & Kolter, J. Z. A simple and effective pruning approach for large language models. In Workshop on Efficient Systems for Foundation Models @ ICML2023 (2023). URL <https://openreview.net/forum?id=tz9JV2PRsv>.
56. Hu, E. J. et al. Lora: Low-rank adaptation of large language models (2021). URL <https://arxiv.org/abs/2106.09685>.
57. Ma, X., Fang, G. & Wang, X. Llm-pruner: On the structural pruning of large language models (2023). 2305.11627.

Reviewer #3 (Remarks to the Author):

In the article the authors present an innovative approach for parsing scientific literature using fine-tuned large language models. This study offers a compelling solution to a complex problem and demonstrates its application in the field of materials science. By utilizing the GPT-3 fine-tuning API, the authors effectively train models to extract information about materials from unstructured text. Their analysis of model performance, comparison of different schemas, and evaluation of a human-in-the-loop approach are particularly interesting. However, there are several revisions that could improve the manuscript:

- 1. The authors should clarify the performance of GPT-3/3.5/4 in few-shot settings. They could select a few of the most challenging examples from the dataset and use them in the prompt with a textual description of the problem.*
- 2. It would be interesting to examine the zero-shot performance in the same setting, using a prompt such as, "extract formulas, descriptions, and applications for every material appearing in the text and present it as a JSON file.*
- ...*
- 4. As the authors mention, open-source language models might be able to reproduce the results demonstrated in this study. Including experiments with such models would be a valuable addition."*

Response to points 1, 2, and 4:

ChatGPT and GPT-4 were not available at the time this work was completed, but we have since performed some of such experiments. However, we found GPT-3.5/4 performance in few- and zero- shot settings are generally poor. We even explored retrieval augmented generation via looking for similar examples from the training set via semantic search over Ada

embeddings and including the training examples in the prompt, but performance was not much improved.

For example, here is the output of one of the more successful (but still representative) examples of using GPT-4 in a few-shot retrieval augmented generation context. Prompt-completion pairs for the 10 most similar abstracts were provided to the model as previous messages and the following instructions were provided as the system message:

```
"""
```

```
Your current task is to extract data from materials science research paper abstracts. Here is the JSON schema you MUST use. Only output the extracted data in this schema. Do not fill in any information that is not explicitly in the abstract. If you don't know something from the context, just leave that spot blank (i.e. don't guess!)
```

```
Make a list of json objects. One for each individual material in the abstract. What is a material? A material is a chemical compound such as "titania", "SiO2", or "graphene". A material is NOT a device (e.g. "valve-regulated lead/acid battery". That would be an application.) For composite materials, make one entry for each part of the composite and put the fact it's a composite (and what composite) as one element in the description.
```

```
SCHEMA:
```

```
[  
  {  
    "name": (string) The material's English name or other identifier (e.g. copper oxide, poly-propylene, BB-1, etc). NOT THE FORMULA OR ACRONYM!,  
    "formula": (string) The chemical formula for the material (e.g. BiSTiO3). NOT THE NAME OR ACRONYM!,  
    "acronym": (string) Any acronym used to refer to a material (e.g. PEO for polyethylene oxide). NOT THE NAME OR FORMULA!,  
    "description": (list of string) Descriptions of the material sample. For example, if it is a thin film, nanoparticles, doped with an impurity, etc.  
    "structure_or_phase": (list of string) Any information that indicates the materials crystal structure or phase (e.g. 'cubic', 'rocksalt', 'HCP', 'alpha-phase', etc), Also extract lattice constants and angles here, if present. (DO NOT put things like "nanoporous" that describe the physical structure. Those are descriptions.)  
    "applications": (list of string) Applications the material is being studied/used for (e.g. 'Li-ion batteries', 'cathode', 'catalyst', etc)  
    If the material is a catalyst, include the reaction it catalyzes.  
  },  
  ...  
]
```

```
"""
```

Abstract:

“Organic–inorganic single-crystalline perovskites have attracted significant attentions due to their exceptional progress in intrinsic properties’ investigation and applications in photovoltaics and optoelectronics. In this study, the large perovskite CH₃NH₃PbI₃ single crystal with the largest length of 80 mm was prepared through the method of inverse-temperature crystallization. Meanwhile, the mass production of integrate photodetectors have been fabricated on the single-crystalline wafer and the photoresponse performances were investigated. The results show that the single-crystalline photodetectors have broad spectrum response to 900 nm, rapid response speed (40 μs) and excellent stability. These findings are of great importance for future promising perovskite single crystalline for integrated photoelectronic application.”

GPT-4 Annotation:

```
[
  {
    "acronym": "",
    "applications": [
      "photovoltaics",
      "optoelectronics",
      "photodetectors"
    ],
    "name": "CH3NH3PbI3",
    "formula": "",
    "structure_or_phase": [
      "perovskite"
    ],
    "description": [
      "single crystal",
      "photodetectors"
    ]
  }
]
```

Human Annotation:

```
[
  {
    "acronym": "",
    "applications": [
      "photovoltaics",
      "optoelectronics",
      "photodetectors",
      "integrated
photoelectronic"
    ],
    "name": "",
    "formula": "CH3NH3PbI3",
    "structure_or_phase": [
      "perovskite"
    ],
    "description": [
      "single-crystalline"
    ]
  }
]
```

For even this extremely easy example, GPT-4 makes basic mistakes that it was specifically warned about in the prompt (mistaking formulas for names, applications for descriptions, repeating an entity in two different categories, etc) *even when multiple highly similar examples were provided as context*.

At this time, we don't think it would be appropriate to include this unsuccessful zero/few-shot work in this paper because adequately describing all of those experiments (variations on prompts, etc) and their negative results in a scientifically rigorous way would require greatly expanding this paper's scope and length without significantly changing its conclusions.

However, as these models improve we expect that few-shot prompting will eventually become a viable strategy for scientific information extraction. We feel that follow-up work on zero/few-shot prompting with LLMs is most likely appropriate for its own paper, perhaps to be combined with comparison to fine-tuning newer open-source LLMs.

The recent work by Zheng et al (<https://doi.org/10.1021/jacs.3c05819>) is a promising avenue in this direction, but their method is focused on (A) creating tabular representations of scientific

data and (B) being a research assistant/knowledge engine - not extracting complex inter-relations between entities.

3. The contents of Tables II and III are somewhat confusing. Table III appears to contain information about entity recognition scores (manual), while Table II focuses on relation extraction (automatic). When comparing the values in the text, the authors use the General-JSON section of Table II. However, it remains unclear what the 0.613 value for the formula score refers to. Additionally, some values (e.g., recall for names) are lower in Table III than in Table II, though the automatic evaluation should provide a lower bound for the scores.

Response:

We thank the reviewer for pointing out this opportunity to clarify the results presented in the two tables in question.

To clarify the difference between exact word-match scoring (Table II) and manual scoring (Table III) in the main text, the paragraph describing the difference between the two tables now reads,

“To account for these factors, we manually scored outputs against the original human (true) annotations for a random 10% test set of the general materials information extraction dataset. We calculated "manual scores" by marking extractions as correct if the core information from entities is extracted in the correct JSON object (i.e., grouped with the correct material formula) and incorrect if they are in the wrong JSON object, are not extracted at all, or are not plausibly inferred from the original abstract. In contrast to the exact match scores (Table II), manual scores allow for flexibility with respect to three aspects: (1) entity normalization, (2) error correction, and (3) multiple plausible annotations of an entity under different labels (e.g., "thermoplastic elastomer" may be considered either an application or description). Whereas Table II assesses whether the model can extract pairs of words *exactly* as they appear in the true annotation, the manual scores shown in Table III assess if the model extracts *equivalent* information to that of the true annotation - regardless of the exact form. Simply, if a domain expert would agree the model's extraction and the true extraction are equivalent, the model's extraction is correct. We provide precise details on this procedure in the Methods section and detailed examples with explanations in the Supplementary Information.”

Table II is our best attempt at a stringent, automatic scoring that does not allow for flexibility in terms. For example, for the sentence “Bismuth telluride is a thermoelectric generator”, this scoring system will look for the exact formula (“bismuth telluride”) linked to the exact application (“thermoelectric generator”). A formula detected as “Bi₂Te₃” linked to the application “thermoelectric generator” will result in score of zero because none of the desired pairwise combinations (“bismuth->thermoelectric”, “bismuth->generator”,

“telluride->thermoelectric” and “telluride->generator”) are exactly detected. Similarly, if the sentence is parsed as a formula of “bismuth telluride” linked to “TE generator”, it will result in only a partially correct score (“bismuth” and “telluride” are correctly linked to “generator”, but considered incorrectly linked to “TE” instead of “thermoelectric”). In contrast, Table III is our best attempt to score in a manner that is consistent with researcher expectations, and would result in a fully correct annotation even if “bismuth telluride” is parsed as “Bi₂Te₃” or “thermoelectric generator” is parsed as “TE generator”).

To further clarify manual scoring, we have included two examples - one simple and one complex - in the supplementary information under “Manual scoring examples”. We include detailed explanations of each example, as well as a full comparison (including individual entities color-coded by true/false positive/negative) for every entry in these examples.

Lower bound: With the issue in the scoring code fixed, the “lower bound” issue should be resolved; all recalls for Table II General-JSON GPT-3 model are now lower than their manually-scored counterparts. However, we should clarify that the manual scores and the exact match scores are independent, and the “lower bound” is not a mathematically precise lower bound. We have amended the main text mentions of lower bound to “approximate” and “rough” lower bounds.

Unclear formula score: The formula entity in Table III simply reflects the NER score of formulae (were all the chemical formulae present in the text correctly detected, without any extraneous detections?) but includes normalization and error correction similarly to the other entities. We note that unlike the other scores, there is no linking component to the formula score. This is because we chose formula as the “root” entity (see Methods), meaning it is the main key under which all under entities are counted as grouped correctly or incorrectly during manual scoring and thus is always correctly “linked” to its root.

REVIEWERS' COMMENTS

Reviewer #1 (Remarks to the Author):

My concerns have been addressed in the revisions. After the comments made by another reviewer, I only think the authors should now cite the article <https://pubs.acs.org/doi/10.1021/acs.jcim.3c00285> that shows the mistakes ChatGPT 3.5 version makes with chemical representations to justify the authors comments to the other reviewer. After including this citation, I agree that the present article should be accepted for publication in Nature Communications.

Reviewer #3 (Remarks to the Author):

The authors have addressed most of my concerns, and clarified various points raised (especially about scoring), which substantially improved the manuscript. Making data and code available is also a great choice and greatly appreciated.

However, in the response regarding zero-shot experiments, the authors say that including the failed experiments would require expanding the paper's scope and length without changing its conclusions significantly. Although this is a reasonable argument, it would still be beneficial for the reader to get a sense of the challenges faced by these models outside of the main conclusions. A suggestion would be to add a short paragraph or section describing these in the Supporting Information and briefly mention it in the paper.

Overall, the revisions and additional experiments have significantly improved the manuscript.

Responses to Reviewer Comments

General response:

We have edited the manuscript to comply with the editors' requests in the submission checklist. This necessitated moving several paragraphs in the main text introduction and restructuring the order of supplementary information. The content is unchanged aside from this reorganization and the addressing of the reviewers' comments.

Reviewer #1:

"My concerns have been addressed in the revisions. After the comments made by another reviewer, I only think the authors should now cite the article <https://pubs.acs.org/doi/10.1021/acs.jcim.3c00285> that shows the mistakes ChatGPT 3.5 version makes with chemical representations to justify the authors comments to the other reviewer. After including this citation, I agree that the present article should be accepted for publication in Nature Communications."

Response:

We thank the reviewer for suggesting this relevant reference. We have included this citation in the prior work section of the introduction, and we discuss their findings further in the Supplementary Information in the context of Reviewer #3's requests. The text in the introduction now reads:

*"Similarly, Castro Nascimento and Pimentel⁴⁶ examined ChatGPT's general knowledge of chemistry; however, they find that, as opposed to methods using considerable prompt engineering⁴⁷, ChatGPT without prompting "tricks" performs poorly on several simple tasks in chemistry. Xie *et al.*'s (2023)⁴⁸ approach utilizes LLMs fine-tuned on a large, broad materials science corpus for a range of Q/A, inverse design, classification, and regression tasks. While these methods^{44,46-48} demonstrate LLMs might act as materials science knowledge engines, they have not been shown to extract structured representations of complex hierarchical entity relationships generalizing *outside* of the pretraining corpus."*

Including the new citations:

46. Castro Nascimento, C. M. & Pimentel, A. S. Do large language models understand chemistry? a conversation with chatgpt. *Journal of Chemical Information and Modeling* 63, 1649–1655 (2023). URL <http://dx.doi.org/10.1021/acs.jcim.3c00285>

47. White, A. D. et al. Assessment of chemistry knowledge in large language models that generate code. *Digital Discovery* 2, 368–376 (2023). URL <http://dx.doi.org/10.1039/D2DD00087C>

Reviewer #3:

“The authors have addressed most of my concerns, and clarified various points raised (especially about scoring), which substantially improved the manuscript. Making data and code available is also a great choice and greatly appreciated.

However, in the response regarding zero-shot experiments, the authors say that including the failed experiments would require expanding the paper's scope and length without changing its conclusions significantly. Although this is a reasonable argument, it would still be beneficial for the reader to get a sense of the challenges faced by these models outside of the main conclusions. A suggestion would be to add a short paragraph or section describing these in the Supporting Information and briefly mention it in the paper.”

Response:

We thank the reviewer for their suggestion. We have included a sentence in the discussion clarifying that zero-shot models may provide an alternative approach to extracting scientific information:

“Similarly, zero-shot approaches without fine-tuning may make scientific information extraction more accessible at the expense of accuracy (see Supplementary Information).”

We discuss our findings further in the supplementary information with several paragraphs of discussion, including two fully explained GPT-4 examples with incorrect

entities/reasons highlighted. We discuss these results in context of similar ChatGPT-based extraction methods (White et al. <https://doi.org/10.1039/D2DD00087C>, Zheng et al. <https://doi.org/10.1021/jacs.3c05819>).